# A Scalable Deterministic Global Optimization Algorithm for Training Optimal Decision Tree

**Kaixun Hua**
University of British Columbia
kaixun.hua@ubc.ca

**Jiayang Ren**
University of British Columbia
rjy12307@mail.ubc.ca

**Yankai Cao**[*]
University of British Columbia
yankai.cao@ubc.ca

## Abstract

The training of optimal decision tree via mixed-integer programming (MIP) has attracted much attention in recent literature. However, for large datasets, state-of-the-art approaches struggle to solve the optimal decision tree training problems to a provable global optimal solution within a reasonable time. In this paper, we reformulate the optimal decision tree training problem as a two-stage optimization problem and propose a tailored reduced-space branch and bound algorithm to train optimal decision tree for the classification tasks with continuous features. We present several structure-exploiting lower and upper bounding methods. The computation of bounds can be decomposed into the solution of many small-scale subproblems and can be naturally parallelized. With these bounding methods, we prove that our algorithm can converge by branching only on variables representing the optimal decision tree structure, which is invariant to the size of datasets. Moreover, we propose a novel sample reduction method that can predetermine the cost of part of samples at each BB node. Combining the sample reduction method with the parallelized bounding strategies, our algorithm can be extremely scalable. Our algorithm can find global optimal solutions on dataset with over 245,000 samples (1000 cores, less than 1% optimality gap, within 2 hours). We test 21 real-world datasets from UCI Repository. The results reveal that for datasets with over 7,000 samples, our algorithm can, on average, improve the training accuracy by 3.6% and testing accuracy by 2.8%, compared to the current state-of-the-art.

## 1 Introduction

Decision tree, as a typical supervised learning algorithm with strong interpretability, has been applied in a wide range of fields. As a rule-based inductive model, a decision tree concludes a series of decision rules to navigate tasks such as classification or regression. The target of the decision tree is to achieve a low prediction error while maintaining the low complexity of the decision tree structure. Therefore, modelling a decision tree can be treated as an optimization problem that minimizes the training error and the tree complexity. However, even constructing an optimal decision tree with binary class is NP-complete [Laurent and Rivest, 1976]. Since the first establishment of the decision tree algorithm by [Morgan and Sonquist, 1963], many heuristic algorithms, such as ID3 [Quinlan, 1986], C4.5 [Quinlan, 1996] and CART [Breiman et al., 1984], are proposed to achieve the optimization target. However, these heuristic algorithms inevitably fall into sub-optimal solutions.

---

[*]corresponding author

36th Conference on Neural Information Processing Systems (NeurIPS 2022).

A recent study based on a comprehensive benchmark shows that global solutions lead to average absolute improvements in test accuracy of 1 - 5% [Bertsimas and Dunn, 2017]. Several formulations and algorithms have been proposed to learn optimal decision trees through mixed integer programming (MIP) [Verwer and Zhang, 2017, 2019, Aghaei et al., 2020, Günlük et al., 2021], itemset mining [Nijssen and Fromont, 2007, Aglin et al., 2020], tailed branch-and-bound and dynamic programming [Hu et al., 2019, Lin et al., 2020, McTavish et al., 2022, Demirović et al., 2022] or SAT [Narodytska et al., 2018, Avellaneda, 2020, Ordyniak and Szeider, 2021, Hu et al., 2020, Shati et al., 2021]. However, a majority of these works can only address binary features. Binary encoding (converting continuous features into binary features) may lead to an enormous feature space, and may also lose information on the relationship between values of numerical features. Some of the aforementioned works have also proposed several approaches, such as bucketization [Verwer and Zhang, 2019], minimum description length principle [Demirović et al., 2022], and boosting tree based column elimination [McTavish et al., 2022], to deal with this problem.

Besides the works using binary-encoded features, the other research direction that focuses on directly addressing numerical features through MIP is rarely explored. Bertsimas and Dunn [2017] propose a seminal work on modelling the optimal classification tree (OCT) as a MIP problem, which directly address continuous features. This approach avoids the drawback of binary encoding and can be extended naturally to the multivariate (oblique) decision tree problem [Bertsimas and Dunn, 2019]. Recently, Zhu et al. [2020] present an SVM-based MIP formulation for learning optimal multivariate decision trees. This work also presents a sample selection method to select a subset of samples and use it to train the decision tree. Although the authors claim that the sample selection method can handle large datasets, the decision tree learned on the subset is not guaranteed to be the optimal one of the original dataset. The above MIP based approaches all rely on off-the-shelf MIP solvers such as CPLEX [Cplex, 2020] and Gurobi [Optimization, 2020], and have reported directly addressing datasets with up to 7,000 samples. Bertsimas and Dunn [2017] mentioned in their article that "datasets with tens of thousands of points were tested, but the time required for our methods to generate high-quality solutions was prohibitive". The main obstacle preventing the solvable problem from scaling to large datasets is that all these solvers implement a classical branch-and-bound (BB) scheme, which needs to branch on all binary variables to guarantee convergence, the dimension of which expands linearly with the size of the dataset.

In this paper, we propose a scalable deterministic global optimization algorithm for training optimal decision tree on classification task with numerical features. The first contribution of our work is that we reformulate the optimal decision tree training as a two-stage optimization problem. The stochastic programming community has made great progress in exploiting the structure of this type of problems [Karuppiah and Grossmann, 2008, Li and Grossmann, 2019]. Recently, Cao and Zavala [2019] prove that for a two-stage problem with continuous second stage variables, branching only on the first stage variables can guarantee the convergence under mild assumptions. Many machine learning problems can be modeled as a two-stage problem and can apply this reduced-space BB method [Hua et al., 2021, Shi et al., 2022]. Although for the decision tree training problem, there are both binary and continuous variables in the second stage (i.e., sample specific variables, such as the predicted class of a sample), we prove that the convergence of our branch-and-bound algorithm can still be guaranteed by only branching on the first-stage variables (i.e., variables that represents the structure of the decision tree), the dimension of which is independent of the number of samples. Secondly, we provide several bounding strategies to construct the lower and upper bound of each BB node. Our new MIP formulation consists of a decomposable structure that can be directly parallelized when solving the lower bounding problem. We also present a tight and efficient MIP formula to obtain a lower bound quickly. Thirdly, we develop a sample reduction method that can predetermine the costs of some samples before the lower bound calculation at each BB node. Such a method can be combined with the lower bound calculation to further speed up the solution process, especially for large-scale datasets. The ablation test (Appendix D) performed on datasets reveals a consistant effectiveness on improving the computation speed of the BB process. Our experiments show that some BB nodes (e.g., when training the dataset BANKNOTE) can even have more than 99% of samples to be determined before computing the lower bound.

With these strategies, our tailored reduced-space BB algorithm can scale well on large datasets. For large datasets, the optimal solution found by our algorithm can on average reduce the optimal cost by 12.5% and improve the training accuracy by 3.6% and testing accuracy by 2.8%, compared to the current state-of-the-art [Bertsimas and Dunn, 2019]. Remarkably, our work is the first that

successfully trains a decision tree to global optimum on a dataset with 245,000 samples (1000 cores, less than 1% optimality gap, within 2 hours). We also improve the test accuracy by 1.7% on a dataset with almost one million samples (928,991 samples, 1000 cores, under 56.2% optimality gap within 4 hours), compared with CART and OCT. We highlight that we also report the optimality gap as a criteria to evaluate the goodness of the optimal solution, which is not presented in the majority of MIP based works. Bertsimas and Dunn [2019] states that the current MIP solvers struggle to provide a reasonable optimality gap for their OCT problem. It is also aligned with our numerical analysis that OCT still retains over 90% optimality gap on average for large datasets. At the same time, our algorithm can provide a practical gap on most of the dataset. Although OCT could find a better solution than CART at the early stage of the solving process, our results reveal that an even better solution could be discovered later and can result in better testing accuracy on large datasets.

In the following discussion of the paper, to avoid the ambiguity, we denote the node for the decision tree as *decision node* and *leaf node* and the node for the branch-and-bound algorithm as *BB node*.

## 2 Optimal Decision Tree with Decomposible Structure

In this work, we focus on training an optimal decision tree for the task of classification. Most part of the formulation presented here follows the work of Bertsimas and Dunn [2019]. However, in our formulation, we do not consider the constraint about the minimum number of samples required in the leaf node. Let $\mathcal{S} = \{1, \cdots, n\}$ be the sample set and $\mathcal{K} = \{1, \cdots, K\}$ be the class set. For a given data set $X = \{\mathbf{x}_s \mid \mathbf{x}_s \in \mathbb{R}^P, s \in \mathcal{S}\}$ with corresponding label set $Y = \{\mathbf{y}_s \mid \mathbf{y}_s \in \{0, 1\}^K, s \in \mathcal{S}\}$, where $P$ is the number of features and $K$ is the number of class. Initially, we scale each feature of the dataset to the range between 0 and 1. We seek to find an optimal decision tree model $F : \mathbb{R}^P \longrightarrow \{0, 1\}^K$ with maximum depth $D$, such that:

$$\mathcal{L}(F) = \sum_{s \in \mathcal{S}} E(F(\mathbf{x}_s), \mathbf{y}_s) + \lambda R(F) \tag{1}$$

is minimized. Here, $E(\cdot)$ measures the misclassification error, $R(\cdot)$ is the regularization function that measure the complexity of the optimal decision tree model and $\lambda$ is the complexity parameter. We denote $t \in \{1, \cdots, T\}$ as the node of the tree, where $T = 2^{D+1} - 1$ is the size (total number of nodes) of the tree. Following the definition of Bertsimas and Dunn [2017], let $p(t) = \lfloor t/2 \rfloor$ be the parent decision node of $t$, $A_L(t)$ as the set of ancestors of $t$ whose left branch is followed on the path from root node to node $t$, and $A_R(t)$ as the set of ancestors of $t$ whose right branch is followed on the path from root node to node $t$. The tree node consists of two sets: *decision node* with $t \in \mathcal{T}_D = \{1, \cdots, \lfloor T/2 \rfloor\}$ and *leaf node* with $t \in \mathcal{T}_L = \{\lfloor T/2 \rfloor + 1, \cdots, T\}$. The training of optimal decision tree can be treated as an optimization problem (Problem 2):

$$\min_{a,b,c,d,z,L} \sum_{s \in \mathcal{S}} \left( \frac{1}{\hat{L}} L_s + \frac{\lambda}{n} \sum_{t \in \mathcal{T}_D} d_t \right) \tag{2a}$$

$$\text{s.t.} \quad \frac{1}{2} \sum_{k \in \mathcal{K}} (y_{sk} + c_{kt} - 2y_{sk}c_{kt}) - L_s \leq 1 - z_{st}, \ \forall t \in \mathcal{T}_L \tag{2b}$$

$$\sum_{k \in \mathcal{K}} c_{kt} = 1, \ \forall t \in \mathcal{T}_L \tag{2c}$$

$$\sum_{t \in \mathcal{T}_L} z_{st} = 1 \tag{2d}$$

$$\mathbf{a}_m^T(\mathbf{x}_s + \epsilon - \epsilon_{min}) + \epsilon_{min} \leq b_m + (1 + \epsilon_{max})(1 - z_{st}), \ \forall m \in A_L(t), t \in \mathcal{T}_L \tag{2e}$$

$$\mathbf{a}_m^T \mathbf{x}_s \geq b_m - (1 - z_{st}), \ \forall m \in A_R(t), \ t \in \mathcal{T}_L \tag{2f}$$

$$\sum_{j=1}^{P} a_{jt} = d_t, \ \forall t \in \mathcal{T}_D, j \in \{1, \cdots, P\} \tag{2g}$$

$$0 \leq b_t \leq d_t, \ \forall t \in \mathcal{T}_D \tag{2h}$$

$$d_t \leq d_{p(t)}, \ \forall t \in \mathcal{T}_D \tag{2i}$$

$$0 \leq L_s \leq 1 \tag{2j}$$

$$a_{jt}, d_t \in \{0,1\}, 0 \leq b_t \leq 1 \quad \forall t \in \mathcal{T}_D, j \in \{1, \cdots, P\} \tag{2k}$$

$$z_{st}, c_{kt} \in \{0,1\}, \quad \forall t \in \mathcal{T}_L \tag{2l}$$

$$s \in \mathcal{S} \tag{2m}$$

Here, variable $a, b, c, d$ describe the structure of decision tree. Specifically, variable $d_t$ on node $t \in \mathcal{T}_D$ determines whether a decision node splits or not. We track such split with variable $\mathbf{a}_t = [a_{1t}, \cdots, a_{Pt}]^T \in \{0,1\}^P$ and $b_t \in [0,1]$. The prediction of each leaf node is controlled by the class indicator $c_{kt} \in \{0,1\}, \forall t \in \mathcal{T}_L, k \in \mathcal{K}$. Variable $z_{st}, \forall t \in \mathcal{T}_L, s \in \mathcal{S}$ represents whether sample $s$ falls into leaf $t$. Variable $L_s \in [0,1]$ represents the loss of sample $s$. $y_{sk} \in \{0,1\}$ is the element of vector $\mathbf{y}_s$ which represents whether sample $s$ is labeled $k$. $\hat{L}$ is the parameter to normalize the misclassification with the baseline accuracy. $\epsilon$, $\epsilon_{max}$, and $\epsilon_{min}$ are constants to maintain the calculation stability of problem 2. Their values are generated following Bertsimas and Dunn [2019]. A detailed explanation of constraints can be found in Appendix A at supplementary materials.

## 3 Reduced-space Branch-and-Bound Algorithm

In this section, we model Problem 2 as a two-stage optimization problem and propose a reduced-space branch-and-bound (BB) algorithm with properly designed lower and upper bounds. Algorithm 1 (presented in Appendix B) elaborates the details of our algorithm. The algorithm starts from the initial feasible region $M_0$. It narrows the gap between the lower and upper bound by partitioning $M_0$ recursively into smaller regions and pruning any regions that are verified to contain no optimal solution. In each iteration with particular sub-region $M$, lower bound $\beta(M)$ and upper bound $\alpha(M)$ are computed. If the gap between $\beta$ and $\alpha$ is smaller than $\epsilon$, Algorithm 1 terminates. Otherwise $M$ is partitioned into two disjoint sub-region with some branching strategies explained in Appendix B.2.

A critical feature of our algorithm is that it can ensure the convergence by only branching on the first stage variables (i.e. $a, b, c, d$). Cao and Zavala [2019] prove that for a two-stage optimization problem, if the second stage variables are continuous, then the BB algorithm can converge by branching only on the first stage variables. Although the decision tree training problem is different since we have both continuous and binary second stage variables, we can still prove its convergence.

**Theorem 1.** *Algorithm 1 converges to the global optimum, in the sense that*

$$\lim_{i \to \infty} \alpha_i = \lim_{i \to \infty} \beta_i = f \tag{3}$$

The proof of Theorem 1 is presented in Appendix B.1.

### 3.1 Two Stage Optimization Problem

Note that variables $a, b, c, d$ describe the structure of decision tree and are the same for all sample, while variables $z$ and $L$ are sample-specific variables and describe the allocation and cost of a specific sample. Therefore, Problem 2 can be reformulated as a two-stage optimization problem in the following form:

$$f(M_0) = \min_{m \in M_0} \sum_{s \in \mathcal{S}} Q_s(m) \tag{4}$$

where we denote $m = (a, b, c, d)$ as all first-stage variables and $Q_s(\cdot)$ represents the optimal value of the second-stage problem:

$$Q_s(m) = \min_{z_s, L_s} \frac{1}{\hat{L}} L_s + \frac{\lambda}{n} \sum_{t \in \mathcal{T}_D} d_t \tag{5a}$$

$$\text{s.t. Constraint } 2b - 2l \tag{5b}$$

The closed set $M_0 := [m^l, m^u]$ denotes the region of the first stage variables. $(\cdot)^l, (\cdot)^u$ represent the lower and upper bound of each variable. In each BB node with a specific partition $M \subseteq M_0$, we solve the problem $f(M) = \min_{m \in M} \sum_{s \in \mathcal{S}} Q_s(m)$.

## 3.2 Lower Bounding Problem

**Basic lower bounding problem:** the problem we solve at each BB node has an implicit constraint. That is, all samples share the same first stage variables (i.e., tree structure). This is called non-anticipativity constraints in the stochastic programming community. We can obtain a basic lower bound by relaxing this constraint and solve the resulting problem:

$$\beta(M) := \min_{m_s \in M} \sum_{s \in \mathcal{S}} Q_s(m_s) \tag{6}$$

such a problem can be easily decomposed into $n$ subproblems $\beta_s(M) := \min_{m \in M} Q_s(m)$ with $\beta(M) := \sum_{s \in \mathcal{S}} \beta_s(M)$. The optimal value of $\beta_s$ can be calculated through Algorithm 2 (presented in Appendix C) by enumerating all possible leaf nodes that sample $s$ can fall into, without the need to solve any optimization problems explicitly. The complexity of Algorithm 2 is $O(|\mathcal{T}_D| + |\mathcal{T}_L|)$. Since the depth of the tree is typically small for the sake of interpretability, the speed of Algorithm 2 can be executed much faster than other methods.

**Theorem 2.** *With given bound $M = [m^l, m^u]$, Algorithm 2 finds all possible leaf nodes that sample $s$ can fall into and gives the global optimal value of $\beta_s(M)$.*

The proof of Theorem 2 is in Appendix C.1.

**Relaxed MIP problem:** One byproduct of Algorithm 2 is $\mathcal{T}_{z_s} \subseteq \mathcal{T}_L$, the set of all possible leaf nodes that sample $s$ can fall into. Figure 1 in Section 4 provides an example on the determination of $\mathcal{T}_{z_s}$. We have $z_{st} = 0, \forall t \notin \mathcal{T}_{z_s}$. For each sample, basic lower bounding method computes the lower bound by comparing the label $y_{sk}$ with the prediction $c_{kt}$ of each leaf node in $t \in \mathcal{T}_{z_s}$. Note here basic lower bounding method relaxes the non-anticipativity constraints on $c$ and allow different values of $c$ for each sample. We can enforce this constraint to generate a tighter lower bound by solving the following MIP problem:

$$\beta^R(M) = \min_{c,L} \sum_{s \in \mathcal{S}} \frac{1}{\hat{L}} L_s + \lambda \sum_{t \in \mathcal{T}_D} d_t^l \tag{7a}$$

$$\text{s.t. } L_s \geq 1 - \sum_{t \in \mathcal{T}_{z_s}} \sum_{k \in \mathcal{K}} y_{sk} c_{kt} \tag{7b}$$

$$\sum_{k \in \mathcal{K}} c_{kt} = 1, \quad \forall t \in \mathcal{T}_L \tag{7c}$$

$$0 \leq L_s \leq 1 \tag{7d}$$

$$c_{kt} \in \{0, 1\}, \quad \forall t \in \mathcal{T}_L, k \in \mathcal{K} \tag{7e}$$

$$s \in \mathcal{S} \tag{7f}$$

where $d^l$ is the lower bound of variable $d$. Since the only number of integer variables in Problem 7 is $c$, the dimension of which is typically small and independent of the number of samples, solving Problem 7 is much easier than the original problem. Because Problem 7 is based on the $\mathcal{T}_{z_s}$ calculated from the basic lower bounding method but enforces non-anticipativity constraints on $c$, we have:

**Proposition 3.** $\beta(M) \leq \beta^R(M) \leq f(M)$

Notice that when the variable $c$ is fixed ($c^l = c^u$ in $M$) at a BB node, $\beta(M) = \beta^R(M)$ holds and we do not need to compute $\beta^R(M)$.

**Group decomposition**: for the basic lower bounding method, we treat each sample as one subproblem. One natural extension is to assign multiple samples to a subproblem. Specifically, we define the group set $\mathcal{G} = \{1, \cdots, G\}$, and we partition $\mathcal{S}$ into $G$ groups $\{\mathcal{S}_1, \cdots, \mathcal{S}_G \mid \bigcup_{g=1}^{G} \mathcal{S}_g = \mathcal{S}$ and $\mathcal{S}_i \cap \mathcal{S}_g = \emptyset, \forall i, g \in \mathcal{G}, i \neq g\}$, then we can obtain a lower bound from:

$$\beta^G(M) := \min_{m_g \in M} \sum_{g \in \mathcal{G}} Q_g(m_g), \tag{8}$$

which can be decomposed into $G$ subproblems with $\beta^G(M) = \sum_{g \in \mathcal{G}} \beta_g^G(M)$ and $\beta_g^G(M) := \min_{m_g \in M} Q_g(m_g)$. Here each subproblem $g$ can be obtained by replacing $s \in \mathcal{S}$ in the original problem with $s \in \mathcal{S}_g$. The non-anticipativity constraints is still enforced within each subproblem (i.e. $a, b, c, d$ are the same for all samples within the group), while the non-anticipativity constraints between groups are relaxed. Hence, the $\beta^G$ provides a tighter lower bound than the basic lower bound $\beta$.

**Proposition 4.** $\beta(M) \leq \beta^G(M) \leq f(M)$

The fewer groups we split, the tighter is the lower bound. However, the solution time of each subproblem is also longer. Therefore, there is a trade-off between the efficiency and tightness when setting the number of groups. Besides this, the way of assigning samples to groups also influences the quality of lower bounds. Finding the best grouping scheme is itself NP-hard. In our implementation, we adopted the heuristic results (e.g. CART) to navigate the assignment of groups. First, we divide all samples into raw groups based on leaves they fall under from the heuristic solution. Second, in each raw group, a $k$-means clustering is launched. Finally, the samples for each cluster at each raw group are evenly assigned to $G$ groups. In this way, samples are approximately evenly distributed among different groups. We keep the group assignment the same for all BB nodes.

### 3.3 Upper Bounding Problem

The upper bound can be easily obtained by choosing any arbitrary first stage solution $\hat{m} \in M$. Let $\alpha(M)$ be the upper bounding problem, we have $\alpha(M) = \sum_{s \in \mathcal{S}} Q_s(\hat{m})$. The calculation of $\alpha(M)$ is the same as the evaluation of a decision tree $\hat{m}$, which is computationally cheap. In our implementation, at the root BB node, we use CART to provide a feasible solution $\hat{m}$ for the initial upper bound. We also propose two heuristics in each descendent BB node to enable a fast search for an optimal solution. First, the optimal solution to each subproblem directly provides a feasible solution. We calculate the corresponding cost and pick the most optimal one to be the upper bound. This method does not require additional computational time to generate $\hat{m}$. However, it could be affected by the group assignment. The second method applies bootstrapping to randomly generate a small subset of samples and train a decision tree for the subset. This "bootstrapped" subproblem provides a feasible solution and introduces more possibilities for finding a better upper bound.

## 4 Sample Reduction

During the calculation of the basic lower bound, Algorithm 2 reduces the possible leaf nodes that sample $s$ can fall into from $\mathcal{T}_L$ to $\mathcal{T}_{z_s}$. Comparing the label $y_{sk}$ with the range of $c$ of each leaf node in $t \in \mathcal{T}_{z_s}$, the loss of some samples can be determined. Specifically, suppose $k$ is the label of sample $s$ (i.e. $y_{sk} = 1$). Equation 9 checks the loss of sample $s$ under $M$.

$$L_s = \begin{cases} 1 & \text{if } \bigwedge_{t \in \mathcal{T}_{z_s}} c_{kt}^l = c_{kt}^u = 0 \\ 0 & \text{if } \bigwedge_{t \in \mathcal{T}_{z_s}} c_{kt}^l = c_{kt}^u = 1 \\ undtm. & \text{otherwise} \end{cases} \tag{9}$$

Figure 1 provides an example to illustrate the determination of $\mathcal{T}_{z_s}$ and the loss determination of a sample $s$ under some conditions of $c^l, c^u$. If the loss of sample $s$ is determined at the current BB node, we no longer need to take this sample into consideration in the lower bound calculation, resulting in sample reduction. Moreover, determined loss holds for all descendent BB nodes. Therefore, the determination of some sample's loss can significantly reduce the computational load, especially when the level of the search tree goes deep. With sample reduction, many of these samples can determine their loss without being involved in the solution of any optimization problem. Indeed, for dataset SKIN-SEGMENTATION (one of the largest test dataset in our experiment with $n = 245,057$), on average, $63.6\%$ of the samples can be determined among all BB nodes. For some BB nodes, $99.6\%$ of samples can have their loss determined. Such a strategy significantly improves node exploration efficiency.

We notice that a similar "data selection" method is proposed in [Zhu et al., 2020]. A major difference between their method and our "cost determination" method is that the solution found from the

"selected" data subset can not guarantee the global optimum of the original dataset. In contrast, in our method, the loss of removed samples is not deleted but pre-determined and added to the final total loss calculation. Hence, the optimal solution we obtain is the same as the optimal solution of the original problem.

**Improvements on the calculation of lower bound:** define the sample set with determined loss as $\mathcal{S}_{dt}$ and those with undetermined loss as $\mathcal{S}_{ud}$, such that $|\mathcal{S}_{dt}| + |\mathcal{S}_{ud}| = n$. Here, we analyze the effects of sample reduction combined with the reach-leaf determination ($\mathcal{T}_{z_s}$) on three lower bound strategies. Initially, the calculation of the basic lower bounding problem requires running Algorithm 2 for $n$ times. With sample reduction, the basic lower bounding problem (Problem 6) can be rewritten as:

$$\beta(M) := \min_{m_s \in M} \sum_{s \in \mathcal{S}_{ud}} Q_s(m_s) + \sum_{s \in \mathcal{S}_{dt}} \frac{1}{\hat{L}} L_s \tag{10}$$

Here, $L_s, \forall s \in \mathcal{S}_{dt}$ is already determined at the parent BB node via the sample reduction. Thus, the computing load of Problem 6 can be reduced from $O(n)$ to $O(|\mathcal{S}_{ud}|)$. Since $M$ of a descendent BB node is a subset of its parent BB node, $\mathcal{T}_{z_s}$ can also be inherited from the parent BB node, and the calculation of $\mathcal{T}_{z_s}$ only need to be executed on samples in $\mathcal{S}_{ud}$. Similarly, for the lower bound from relaxed MIP, the number of samples involved in the optimization is reduced from $n$ to $|S_{ud}|$. Finally, for the group decomposition based lower bounding strategy, the number of samples involved in each subproblem is reduced from $|\mathcal{S}_g|$ to $|\mathcal{S}_{ud} \cap \mathcal{S}_g|$. The detailed formulations of these lower bounding methods after sample reduction can be found in Appendix D.

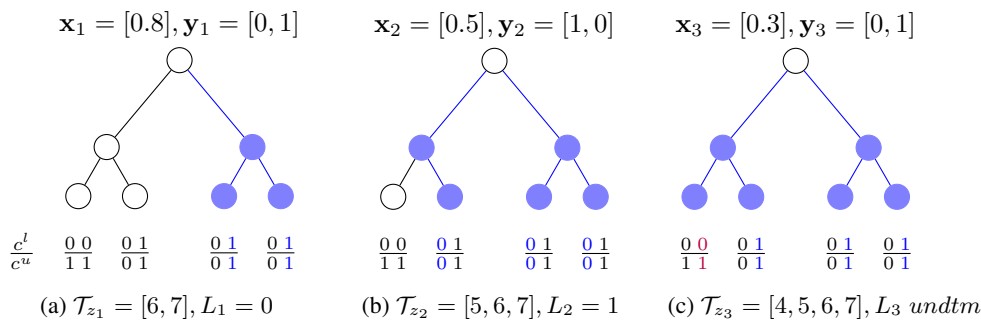

(a) $\mathcal{T}_{z_1} = [6, 7], L_1 = 0$     (b) $\mathcal{T}_{z_2} = [5, 6, 7], L_2 = 1$     (c) $\mathcal{T}_{z_3} = [4, 5, 6, 7], L_3\ undtm$

Figure 1: Loss determination under the bounds $a^l = [1, 1, 1], a^u = [1, 1, 1], b^l = [0.2, 0.1, 0.1], b^u = [0.6, 0.4, 1.0], c^l = \begin{bmatrix} 0 & 0 & 0 & 0 \\ 0 & 1 & 1 & 1 \end{bmatrix}, c^u = \begin{bmatrix} 1 & 0 & 0 & 0 \\ 1 & 1 & 1 & 1 \end{bmatrix}$. The nodes in blue indicate the possible paths of a sample under current bound. The bound of $c$ for each leaf $t$ is denoted as $\frac{c^l}{c^u}$ under each leaf node. **Sample** 1: since $\mathbf{x}_1[1] \geq b_1^u$, $\mathbf{x}_1$ can only fall into the right descendent of the root decision node, and as $\mathbf{x}_1[1] \in [b_3^l, b_3^u]$, leaf node 3 and 4 can be reached by sample 1 and thus $\mathcal{T}_{z_1} = \{6, 7\}$. Since $c_{2t}^l = c_{2t}^u = 1, \forall t \in \mathcal{T}_{z_1}$ (colored in blue), which is identical to $y_{12}$, we have $L_1 = 0$. **Sample** 2: since $\mathbf{x}_2[1] \in [b_1^l, b_1^u]$, $\mathbf{x}_2$ can fall into both descendent of the root decision node, However, as $\mathbf{x}_2[1] \geq b_2^u$, leaf node 1 can not be reached by sample 2, and thus $\mathcal{T}_{z_2} = \{5, 6, 7\}$. Since $y_2$ is inconsistent with all the $[c_t^l, c_t^u], \forall t \in \mathcal{T}_{z_2}$, we determine that $L_2 = 1$. **Sample** 3: since $\mathbf{x}_3[1] \in [b_t^l, b_t^u], \forall t \in \mathcal{T}_D$, all leaf node can be reached by sample 3, $\mathcal{T}_{z_3} = \{4, 5, 6, 7\}$. Sample 3 can have several possible loss values if fall into different leaf node in $\mathcal{T}_{z_3}$. Here, variable $c$ is fixed in all leaves except leaf 1 (colored in red). Therefore, we are unable to determine $L_3$.

## 5 Computational Experiments

We evaluate the performance of our algorithm (denoted as RS-OCT) on 21 real-world datasets from UCI machine learning repository [Lichman et al., 2013]. The algorithm is implemented in Julia 1.7.0. Our implementation code is available at: `https://github.com/YankaiGroup/optimal_decision_tree`. Comparison is made against the CART algorithm [Breiman et al., 1984] and Optimal Classification Tree (OCT) method [Bertsimas and Dunn, 2017]. A comparison with the performance of FlowOCT [Aghaei et al., 2020] is also presented in Appendix E.1. To avoid the effects of hyper-parameter choice, the penalty parameter $\lambda$ is set to $0.05$ for all three algorithms. The split of training and testing data follows the work of Bertsimas and Dunn [2017], that is, $75\%$ of the entire dataset is selected as the training set while the rest $25\%$ is set as the testing set. All datasets are

split by the `stratifiedobs` function from Julia Package `MLDataUtils`, under random seed 1, so that the distribution of class is guaranteed to be the same in both training and testing set. RS-OCT and OCT are executed until either they converge to an optimality gap within $1\%$ or the runtime exceeds a limit of 4 hours. The result of CART is obtained through the Julia Package `DecisionTree` with version `v0.10.12`. The MIP problem of OCT and the subproblem of RS-OCT (for grouping based lower bound) are solved by the state-of-art global optimizer CPLEX 20.1.0 [Cplex, 2020]. Both OCT and RS-OCT are provided with the same CART solution as warm-start. A preliminary parallel version of our algorithm is implemented by assigning subproblems in the lower bound calculation to multiple CPU cores. The size of each subproblem for group decomposition is set as $max(2K, \frac{2^D n}{(ct+P+2)-2^D(P+2+K)})$, where $ct$ (default: 150) is a constant that controls the average group-size. The serial and parallel experiments are executed on a high-performance computing cluster (40 Intel "Skylake" at 2.4 GHz with 202G RAM per computing node). Five criteria (upper bound, training accuracy, testing accuracy, optimality gap, and time) are used to compare the performance of the algorithms. Here, the upper bound (UB) represents the best feasible solution we found until the termination of each algorithm. The optimality gap is calculated as $\frac{UB-LB}{UB} \times 100\%$. It measures the closeness from the best-searched solution (i.e. UB) to its possible global optimum (i.e. lower bound). As a unique property of deterministic global optimization algorithms, heuristic algorithms like CART cannot provide such criteria. Our work focuses on improving the solution process of the optimal decision tree problem. Therefore, to best represent the solving performance, we use the same value of $\lambda$, so that both our method and OCT are solving the same optimization problem, and we can have a fair comparison of the optimality gap, solution time, and best achieved upper bound on this specific problem formulation. However, to further demonstrate the performance of RS-OCT in practical, compared to other state-of-the-art algorithms, we also perform experiments with hyperparameter tuning on depth 2 and 3. The results are presented in Appendix E.1. The effectiveness of sample reduction is validated via the ablation test and the result is shown in Appendix E.2.

**Small and medium-scale datasets** We perform experiments on 15 datasets ranging from 210 to 5,473 samples. Table 1 presents the result of using three algorithms. In terms of UB, RS-OCT can find a better (or equally better) solution with the lowest cost for almost all datasets, compared to CART and OCT. RS-OCT also terminates with a smaller optimality gap for all datasets. It is remarkable that for datasets such as GLASS, BANKNOTE, and WALL-FOLLOWING, RS-OCT can achieve an optimality gap within $1\%$ while OCT still has a large gap over $98\%$ after 4 hours of running. Our result indicates that although OCT could find a better solution than CART, it is still hard to claim that such a solution is the optimal one. In contrast, RS-OCT can find an even better solution more efficiently and prove the optimality within a reasonable time. Due to the better cost found by RS-OCT, the training accuracy of all datasets in Table 1 improves by $2.9\%$ on average compared to CART and $0.8\%$ on average compared to OCT. In terms of testing accuracy, since we fixed the penalty parameter $\lambda$ for all datasets, the result on smaller datasets ($n < 1000$) is unsatisfactory probably due to an overfitting issue. In total, the testing accuracy is improved by $2.2\%$ and $1.0\%$, on average, compared to CART and OCT, respectively. For medium-scale datasets ($n \geq 1000$), the testing accuracy improves by $3.3\%$ on average compared to CART and $1.4\%$ on average compared to OCT.

**Large-scale datasets** We test our algorithm on several large datasets ($n \geq 7,000$), with a variety of dimensions (3-16) and number of classes (2-12), as shown in Table 2. RS-OCT is executed in parallel with 1000 CPU cores. Remarkably, RS-OCT outperforms OCT and CART on all criteria for all datasets, including the HT-SENSOR which has almost one million samples. In terms of upper bound, we notice that for datasets with a size over $40,000$ samples, OCT can not provide a better solution than the warm-start given by CART. Nevertheless, RS-OCT discovers better solutions in terms of UB than both CART ($12.5\%$ improvement on average) and OCT ($11.7\%$ improvement on average) on all large datasets. In terms of optimality gap, RS-OCT converges to an optimality gap within $1\%$ on PENDIGITS, SHUTTLE, and SKIN-SEGMENTATION. It verifies that the best solutions found for these datasets are merely at most $1\%$ close to the actual global optimal solution. It is the first time an optimal decision tree (without binary encoding) is found for a dataset with almost **one million** samples with a meaningful optimality gap ($56.2\%$) in four hours of runtime. Remarkably, our algorithm can also solve for a dataset with over $200,000$ samples to a practical optimality gap ($\leq 1\%$) in less than two hours. Given the more optimal solution found by RS-OCT, its training accuracy is on average boosted by $3.7\%$ and $3.6\%$, compared to CART and OCT, respectively. The testing accuracy

Table 1: Numerical results on small and median datasets ($D = 2$, $\lambda = 0.05$, serial). The results of testing accuracy reported in [Bertsimas and Dunn, 2017] are also listed in the paranthesis for CART and OCT. These "paranthesis" results are obtained within the runtime of 2 hours.

| DATA-SET | $n$ | $P$ | $K$ | METHOD | UB | TRAINING ACCURACY(%) | TESTING ACCURACY(%) | GAP (%) | TIME (S) |
|---|---|---|---|---|---|---|---|---|---|
| SEEDS | 210 | 7 | 3 | CART | 42.2 | 91.2 | **94.1** (87.2) | - | - |
| | | | | OCT | **27.2** | **94.3** | 86.3 (88.7) | <1% | 579.1 |
| | | | | RS-OCT | **27.2** | **94.3** | 88.2 | <1% | **151.2** |
| GLASS | 214 | 9 | 6 | CART | 152.2 | 65.8 | **64.3** | - | - |
| | | | | OCT | 143,8 | 67.7 | **64.3** | 99.9 | 14400 |
| | | | | RS-OCT | **140.9** | **68.4** | 62.5 | **<1%** | **2503.8** |
| BODY | 507 | 5 | 2 | CART | 74.3 | 90.0 | 88.9 | - | - |
| | | | | OCT | **58.7** | **92.1** | **92.1** | <1% | 10985 |
| | | | | RS-OCT | **58.7** | **92.1** | **92.1** | <1% | **965.9** |
| STATLOG-GERMAN | 1,000 | 24 | 2 | CART | 281.6 | 73.7 | 71.2 (70.1) | - | - |
| | | | | OCT | 280.2 | 73.9 | 71.2 (70.5) | 99.9 | 14400 |
| | | | | RS-OCT | **267.3** | **75.1** | **72.0** | **38.5** | 14400 |
| CONCRETE | 1,030 | 8 | 3 | CART | 667.6 | 62.6 | 61.9 | - | - |
| | | | | OCT | **621.4** | **65.2** | **65.4** | 100 | 14400 |
| | | | | RS-OCT | **621.4** | **65.2** | 63.0 | **13.0** | 14400 |
| BANKNOTE | 1,372 | 4 | 2 | CART | 185.6 | 90.0 | 89.0 (89.0) | - | - |
| | | | | OCT | 155.0 | 91.6 | 91.0 (90.1) | 98.7 | 14400 |
| | | | | RS-OCT | **135.2** | **92.7** | **92.2** | **<1%** | **675.6** |
| CONTRA-CEPTIVE | 1,473 | 11 | 3 | CART | 1353.7 | 47.6 | 46.3 (46.8) | - | - |
| | | | | OCT | 1215.6 | 53.0 | 51.0 (48.4) | 54.5 | 14400 |
| | | | | RS-OCT | **1189.8** | **54.0** | **56.4** | **12.0** | 14400 |
| OZONE-EIGHT | 1,847 | 72 | 2 | CART | 103.3 | 93.1 | **93.1** (93.1) | - | - |
| | | | | OCT | 102.2 | 93.2 | 92.8 (93.1) | 100 | 14400 |
| | | | | RS-OCT | **94.7** | **93.7** | 92.4 | **97.6** | 14400 |
| OZONE-ONE | 1,848 | 72 | 2 | CART | 43.5 | 97.0 | 96.8 (96.8) | - | - |
| | | | | OCT | **39.3** | **97.3** | 96.8 (96.8) | 99.9 | 14400 |
| | | | | RS-OCT | 40.4 | 97.2 | 96.8 | **99.6** | 14400 |
| THYROID-ANN | 3,772 | 21 | 3 | CART | 140.6 | 95.4 | 95.6 (95.6) | - | - |
| | | | | OCT | 86.7 | 97.2 | 96.8 (95.6) | 99.9 | 14400 |
| | | | | RS-OCT | **66.1** | **97.8** | **98.1** | **65.4** | 14400 |
| STATLOG-LANSAT | 4,435 | 36 | 6 | CART | 4861.3 | 64.7 | 64.6 (63.2) | - | - |
| | | | | OCT | 4786.8 | 65.2 | 64.8 (63.2) | 100 | 14400 |
| | | | | RS-OCT | **4757.8** | **65.4** | **66.0** | **40.1** | 14400 |
| SPAMBASE | 4,601 | 57 | 2 | CART | 826.9 | 85.5 | 84.5 (84.2) | - | - |
| | | | | OCT | 826.9 | 85.5 | 84.5 (84.3) | 100 | 14400 |
| | | | | RS-OCT | **756.0** | **86.7** | **85.9** | **81.4** | 14400 |
| WALL-FOLLOWING | 5,456 | 2 | 4 | CART | 608.8 | 94.0 | 94.0 (94.0) | - | - |
| | | | | OCT | 608.8 | 94.0 | 94.0 (94.0) | 100 | 14400 |
| | | | | RS-OCT | 608.8 | 94.0 | 94.0 | **<1%** | **223.2** |
| PAGE-BLOCK | 5,473 | 10 | 5 | CART | 315.4 | 93.1 | 93.2 | - | - |
| | | | | OCT | 273.1 | 93.9 | 93.8 | 100 | 14400 |
| | | | | RS-OCT | **209.6** | **95.4** | **95.0** | **46.2** | 14400 |

is also boosted by 3.3% and 2.8% on CART and OCT, respectively. Notably, compared with the results from Table 1, we find that the performance of RS-OCT is improving with the increment of the data size.

## 6 Conclusion

This paper proposed a tailed reduced-space branch and bound (BB) algorithm to train optimal decision tree for the classification tasks. Our BB algorithm can converge to a global $\epsilon$-optimal solution by only branching on variables describing the tree structure, which is invariant to the number of samples. Combined with the sample reduction and paralleled lower bound calculation, our algorithm can perform well on all datasets in terms of UB, optimality gap and training accuracy. Our algorithm also improves the testing accuracy on all large datasets ($n \geq 7,000$).

**Impact and Future work:** We hope that the MIP-based scalable framework proposed in this paper can help address the needs of optimal decision tree for large-scale data training in high-stake domains such as crime analysis [Zhuang et al., 2017], or medical decision [Bertsimas et al., 2018]. We look forward to integrating our algorithms with other functionality (e.g., fairness [Aghaei et al., 2019]) to provide handy tools for wider applications.

Table 2: Numerical results on large datasets ($D = 2, \lambda = 0.05$, paralleled with 1000 cores). Bertsimas and Dunn [2017] do not report results on datasets of this scale.

| DATA-SET | $n$ | $P$ | $K$ | METHOD | UB | TRAINING ACCURACY(%) | TESTING ACCURACY(%) | GAP (%) | TIME (s) |
|---|---|---|---|---|---|---|---|---|---|
| PENDIGITS | 7,494 | 16 | 10 | CART | 32993.0 | 38.9 | 39.1 | - | - |
| | | | | OCT | 32993.0 | 38.9 | 39.1 | 100 | 14400 |
| | | | | RS-OCT | **32426.1** | **40.0** | **39.9** | **<1%** | **2093.5** |
| AVILA | 10,430 | 10 | 12 | CART | 9454.3 | 50.4 | 51.0 | - | - |
| | | | | OCT | 9196.4 | 51.7 | 52.5 | 100 | 14400 |
| | | | | RS-OCT | **8787.5** | **53.8** | **53.8** | 19.6 | 14400 |
| EEG | 14,980 | 14 | 2 | CART | 7815.8 | 61.7 | 61.2 | - | - |
| | | | | OCT | 7748.7 | 61.7 | 61.0 | 100 | 14400 |
| | | | | RS-OCT | **6752.7** | **66.9** | **65.5** | 50.8 | 14400 |
| HTRU | 17,898 | 8 | 2 | CART | 327.0 | 97.8 | 97.7 | - | - |
| | | | | OCT | 320.5 | 97.8 | 97.7 | 100 | 14400 |
| | | | | RS-OCT | **300.7** | **98.0** | **97.8** | 42.8 | 14400 |
| SHUTTLE | 43,500 | 9 | 7 | CART | 2567.5 | 93.8 | 93.8 | - | - |
| | | | | OCT | 2567.5 | 93.8 | 93.8 | 100 | 14400 |
| | | | | RS-OCT | **1908.1** | **95.4** | **95.5** | **<1%** | **586.1** |
| SKIN-SEGMEN-TATION | 245,057 | 3 | 2 | CART | 23409.5 | 89.9 | 89.8 | - | - |
| | | | | OCT | 23409.5 | 89.9 | 89.8 | 100 | 14400 |
| | | | | RS-OCT | **16953.7** | **92.7** | **92.7** | **<1%** | **6698.9** |
| HT-SENSOR | 928,991 | 11 | 3 | CART | 782046.1 | 58.1 | 58.1 | - | - |
| | | | | OCT | 782046.1 | 58.1 | 58.1 | 100 | 14400 |
| | | | | RS-OCT | **753075.8** | **59.7** | **59.7** | 56.2 | 14400 |

**Limitation:** Similar to other works, our algorithm only tests optimal decision tree problems with small depth (e.g. $D = 2, 3$). Larger $D$ slows down the algorithm's efficiency. Proper bound tightening methods will be designed in the future to solve this problem.

## Acknowledgments and Disclosure of Funding

Y.C. acknowledges funding from the discovery program of the Natural Science and Engineering Research Council of Canada under grant RGPIN-2019-05499. The authors also gratefully acknowledge the computing resources provided by SciNet (www.scinethpc.ca) and Digital Research Alliance of Canada (www.alliancecan.ca).

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
