# Supplementary Materials

## A Constraint Explanation for Problem 2

Constraint 2b checks the loss of each sample and the derivation is shown as follows.

Let a latent variable $L_{st}$ to be the loss of sample $s$ at leaf node $t$, as $y_{sk}$ and $c_{kt}$ are both binary, we can set:

$$L_{st} = \frac{1}{2} \sum_{k \in \mathcal{K}} (y_{sk} - c_{kt})^2 \tag{11a}$$

$$= \frac{1}{2} \sum_{k \in \mathcal{K}} (y_{sk} + c_{kt} - 2y_{sk}c_{kt}) , \quad \forall t \in \mathcal{T}_L, s \in \{1, \cdots, n\} \tag{11b}$$

It is easy to see that $0 \leq L_{st} \leq 1$, as the loss of each sample $s$ at leaf $t$ can only have one mis-match on class label. We introduce variable $L_s \in [0,1]$ to indicate the loss of sample $s$ in its best allocated leaf. To guarantee that the best leaf node for sample $s$ is chosen, we conclude the following big-M constraints:

$$-M(1 - z_{st}) \leq L_s - L_{st} \tag{12}$$

As $L_s - L_{st}$ will no exceed 1, combined with Equation 11, we have:

$$\frac{1}{2} \sum_{k \in \mathcal{K}} (y_{sk} + c_{kt} - 2y_{sk}c_{kt}) - L_s \leq 1 - z_{st} , \quad \forall t \in \mathcal{T}_L \tag{13}$$

Constraint 2c ensures that only one label should be assigned to a particular leaf.

Constraint 2d to 2i are mainly adopted from Bertsimas and Dunn [2019] on Chapter 8.2. Here we briefly explain the meaning and derivations of these constraints. Constraint 2d guarantees that a sample can only be assigned to one leaf node. Constraint 2e and 2f are set to enforce the split of the tree with given training data. They are derived as follow:

First, to enforce the splits required by the structure of the tree when assigning points to leaves, we set constraints:

$$\mathbf{a}_m^T(\mathbf{x}_s + \epsilon) \leq b_m + M_1(1 - z_{st}), \forall m \in A_L(t), t \in \mathcal{T}_L \tag{14}$$

$$\mathbf{a}_m^T \mathbf{x}_s \geq b_m - M_2(1 - z_{st}), \forall m \in A_R(t), \ t \in \mathcal{T}_L \tag{15}$$

Here $\epsilon$ is added to avoid the strict inequality and can be expressed as the smallest non-zero distance between adjacent values of $j$th feature:

$$\epsilon_j = min\{x_j^{(i+1)} - x_j^{(i)} | x_j^{(i+1)} \neq x_j^{(i)}\} \tag{16}$$

where $x_j^{(i)}$ is the $i$th largest value in the $j$th feature.

To specify the $M_1$ and $M_2$, let $\epsilon_{min} = min_j\{\epsilon_j\}$ and $\epsilon_{max} = max_j\{\epsilon_j\}$, we notice that the largest value for $\mathbf{a}_m^T(\mathbf{x}_s + \epsilon) - b_m$ is $1 + \epsilon_{max}$. Thus, $M_1$ can be set to $1 + \epsilon_{max}$. Similarly, The largest value of $\mathbf{a}_m^T \mathbf{x}_s - b_m$ is 1, so we can set $M_2 = 1$.

Therefore, we reformulate Equation 14 and 15 as follow:

$$\mathbf{a}_m^T(\mathbf{x}_s + \epsilon - \epsilon_{min}) + \epsilon_{min} \leq b_m + (1 + \epsilon_{max})(1 - z_{st}), \forall m \in A_L(t), t \in \mathcal{T}_L \tag{17}$$

$$\mathbf{a}_m^T \mathbf{x}_s \geq b_m - (1 - z_{st}), \forall m \in A_R(t), \ t \in \mathcal{T}_L \tag{18}$$

Constraint 2g and 2h are set to ensure that if there's no split on $t \in \mathcal{T}_D$, then all samples at this node will go to the right branch of this decision node and send down to the same right most leaf node (along with Constraint 2e and 2f). Constraint 2i enforces the hierarchical structure of the tree. It guarantees a parent node must have a split if its child node has a split.

# B Branch-and Bound Algorithm for Optimal Decision Tree Problem

Algorithm 1 depicts the details of the Branch-and-bound scheme for training the optimal decision tree. Based on Proposition 3 and 4, relaxed MIP and group decomposition provide tighter bounds than the basic lower bound. Therefore, using the basic lower and upper bound is adequate to deduct the convergence proof of Algorithm 1. The critical property of our BB based optimal decision tree training algorithm is that it can guarantee the convergence by only branching on the space of first stage variables $m = (a, b, c, d)$ (i.e. the variables describing the tree structure).

Algorithm 1 can be treated as a rooted BB tree. It starts from a root BB node indexed at level 0 with the original variable space $M_0$. Let $\delta(M) = ||m^u - m^l||_\infty$ be the diameter of the set $M$. We denote $M_{i_q}$ as the BB node at level $q$ which is explored at iteration $i_q$. A node $M_{i_{q+1}}$ is a descendent node that connected to its parent node $M_{i_q}$, with $M_{i_{q+1}} \subset M_{i_q}$. The descendent node is at level $q + 1$ and is explored at iteration $i_{q+1}$. We denote $\{M_{i_q}\}$ as the sequence of the partition element that represents a path of the BB tree from the root node to the node $M_{i_q}$ at the level $q$. Since the search space is narrowing along the path, the sequence $\{\beta_i\}$ is monotonically increasing, while $\{\alpha_i\}$ is monotonically decreasing. If $\lim_{i \to \infty} \alpha_i = \lim_{i \to \infty} \beta_i = f$, we say the BB algorithm converges. Once the algorithm is convergent, it produces a global $\epsilon$-optimal solution in a finite number of steps.

---

**Algorithm 1:** Branch-and-Bound Optimal Decision Tree Algorithm

---

**Input:** $M_0$, non-zero tolerance $\epsilon$
1 Set iteration index $i = 0$, $\mathbb{M} \leftarrow \{M_0\}$ ;
2 Initial upper and lower bounds $\alpha_i = \alpha(M_0)$, $\beta_i = \beta(M_0)$;
3 **repeat**
4    **Node Selection**
5      Select a set $M \in \mathbb{M}$ satisfying $\beta(M) = \beta_i$;
6      $\mathbb{M} \leftarrow \mathbb{M} \setminus \{M\}$;
7      $i \leftarrow i + 1$;
8    **Branching**
9      Partition $M$ into subsets $M_1$ and $M_2$ according to the branch strategy in Section B.2;
10      Add each subset to $\mathbb{M}$ to create separated descendent nodes;
11    **Bounding**
12      Compute $\alpha(M_1), \beta(M_1), \alpha(M_2), \beta(M_2)$;
13      For any $j \in \{1, 2\}$, if $\beta_s(M_j)$ is infeasible for some $s \in \mathcal{S}$, $\mathbb{M} \leftarrow \mathbb{M} \setminus \{M_j\}$;
14      $\beta_i \leftarrow \min\{\beta(M') \mid M' \in \mathbb{M}\}$;
15      $\alpha_i \leftarrow \min\{\alpha_{i-1}, \alpha(M_1), \alpha(M_2)\}$;
16      Remove all $M'$ from $\mathbb{M}$ if $\beta(M') \geq \alpha_i$;
17      If $|\beta_i - \alpha_i| \leq \epsilon$, STOP;
18 **until** $\mathbb{M} = \emptyset$;

---

## B.1 Proof of Theorem 1 (Convergence Analysis)

The proof follows the work of Cao and Zavala [2019] which proves the convergence of the BB algorithm by only branching on first stage variables. In their case, both first stage and second stage variables are continuous variables. They also assume that $Q(.)$ is a continous function. However, we prove that for optimal decision tree problem, which has mixed-integer first and second stage variables and also discontinous $Q(.)$ function, such convergence can still hold. We also adopt some basic results from the seminal work in the Chapter IV of Horst and Tuy [2013] and modified definitions and theories accordingly for the optimal decision tree problem.

Specifically, based on Corollary IV.1 and Definition IV.10 from Horst and Tuy [2013], it is easy to see that Algorithm 1 creates *exhaustive subdivisions*. That is, for all decreasing sub-sequences $\{M_{i_q}\}$, $\lim_{q \to \infty} \delta(M_{i_q}) = 0$. To prove that the lower bound $\beta$ converges to $f$, we have to show that the lower bounding operation of Algorithm 1 is *strongly consistent* and its selection operation is *bound improving*.

From Definition IV.6 and Theorem IV.3 of Horst and Tuy [2013], it is trivial that the selection operation in Algorithm 1 is *bound improving*, since it always selects the node with the actual attained

lower bound for further partition. In Lemma 3 of Cao and Zavala [2019], they prove that the basic lower bounding operation is *strongly consistent* if the first stage variables are continuous. In our case, by giving priority to the first stage integer variables, after finite steps, all integer variables are determined and then the prove can be down following Cao and Zavala [2019].

**Lemma 5.** *Given an exhaustive subdivision on $m$, Algorithm 1 satisfies $\lim_{i \to \infty} \beta_i = f$.*

*Proof.* Since Algorithm 1 uses a *strongly consistent* lower bounding operation and its selection operation is *bound improving*, from Theorem IV.3 in Horst and Tuy [2013], the result is trivial. □

Then, we have to prove the convergence of the upper bounds $\alpha$ through the following lemma.

**Lemma 6.** *Given an exhaustive subdivision on $m$, Algorithm 1 generates a sequence $\{\alpha_i\}$ such that $\lim_{i \to \infty} \alpha_i = f$.*

*Proof.* Since $a, c, d$ are binary variables and these variables are given priority in the branching strategy, the value of $a, c, d$ will eventually be fixed after finite step of subdivision. Suppose the optimal limit points of them are $a^*, c^*, d^*$. Therefore, we can have a $i_0$, such that $m' = (a^*, b, c^*, d^*), \forall m' \in M_i, \forall i \geq i_0$ with fixed $a^*, c^*, d^*$. Let $m^* \in M_0$ as an optimal solution of Problem 2, with $m^* = (a^*, b^*, c^*, d^*)$. Without loss of generality, we assume at a decision node $t$, the $j$th feature is selected for splitting (i.e. $a_{jt}^* = 1$). Suppose the sorted value of $j$th feature from small to large is $\{x_{1,j}, \cdots, x_{s,j}, \cdots, x_{n,j}\}$. Assume $x_{s,j}$ is an optimal solution of $b_t$. It is easy to see that all values within the range $[x_{s,j}, x_{s+1,j})$ gives the same optimal value. Define $b_t^*$ as the midpoint of the range. Based on the definition of $\epsilon$ in Problem 2 (See Chapter 8.2 of Bertsimas and Dunn [2019]), we are guarantee to have the same optimal value on any $b$ such that $b_t \in [b_t^* - 0.5\epsilon_{min}, b_t^* + 0.5\epsilon_{min}], \forall t \in \mathcal{T}_D$. We denote $B(m^*) = \{(a^*, b, c^*, d^*) \mid b_t \in [b_t^* - 0.5\epsilon_{min}, b_t^* + 0.5\epsilon_{min}], \forall t \in \mathcal{T}_D\}$. For every point $m \in B(m^*)$, we have $Q(m) = Q(m^*)$ holds.

Since the subdivision is exhaustive, after a finite number of iterations $i'$ such that $i' \geq i_0$, we have, either the partition considered satisfies $M_{i'} \subseteq B(m^*)$, or the partition $M_{i'}$ which contain the solution $m^*$ is pruned. For the first case, since $M_{i'} \subseteq B(m^*)$, we have $Q(m) = Q(m^*), \forall m \in M_{i'}$. Then we have that $\alpha_{i'} = Q(m^*) = f$. In the second one, since $M_{i'}$ is pruned, then we have $\alpha_{i'} \leq \beta(M_{i'})$. Because $m^* \in M_{i'}$, we also have $\beta(M_{i'}) \leq Q(m^*)$. Hence, $\alpha_{i'} = Q(m^*) = f$. Therefore, $\lim_{i \to \infty} \alpha_i = f$ holds. □

Combing Lemma 5 and 6, Theorem 1 can be obtained.

## B.2 Branching Strategy for Algorithm 1

The branching strategy for Algorithm 1 is designed based on the variables' effect on the structure of the optimal decision tree. Since $d$ indicates whether each decision node is splitting or not, it directly determines the skeleton of the tree structure. Thus, in implementation, the variables of $d$ on each decision node are first used for branching. We devise the following heuristics for the other three decision variables $(a, b, c)$ to select the branching variable: First, we generate a threshold $\tau = 1 - \frac{1}{2}||b^u - b^l||_\infty$. Suppose $\tau$ is smaller than a randomly generated number between 0 and 1. In that case, we branch on variable $a$, which determines the splitting feature of the decision node (If $a$ is all determined, then branching on variable $c$). When $\tau$ is larger than the randomly generated number, we branch on variable $b$ (continuous) at the midpoint of its range. The interior order of each type of variables is based on picking variable with smaller decision(leaf) node index (e.g. $a_{j,t}$ will be chosen over $a_{j,t+1}$, if they both are not determined.)

## C  Basic Lower Bounding Problem Calculation

Algorithm 2 explicit the way of finding optimal value of the subproblem $\beta_s$. It also provides a byproduct $\mathcal{T}_{z_s}$, which indicates the leaf node that sample $s$ can reach under region $M$. $\mathcal{T}_{z_s}$ can inherit successively to the descendent BB node and directly reduce the number of variables on $z$ for the lower bound calculation. Moreover, $\mathcal{T}_{z_s}$ can also enhance the effect of sample reduction, which further improves the performance of the BB algorithm.

**Algorithm 2:** Global search for the basic subproblem $\beta_s(M) := \min\limits_{m \in M} Q_s(m)$

---

**Input:** $\mathbf{x}_s, \mathbf{y}_s, m^l = (a^l, b^l, c^l, d^l), m^u = (a^u, b^u, c^u, d^u), D, n, \hat{L}, \lambda$

1  Set DT root node (decision node) $t_0 = 1$, node set $\mathbb{T} \leftarrow \{t_0\}$, Leaf loss set $\mathbb{L} = \emptyset$, reach leaf node set $\mathcal{T}_{z_s} = \emptyset$, $\mathcal{T}_D = \{1, \cdots, 2^D - 1\}$, $\mathcal{T}_L = \{2^D, \cdots, 2^{D+1} - 1\}$;
2  **while** $\mathbb{T} \neq \emptyset$ **do**
3     $t = \min\{t \mid t \in \mathbb{T}\}$;
4     $\mathbb{T} \leftarrow \mathbb{T} \setminus \{t\}$;
5     **if** $t \in \mathcal{T}_D$ **then**
6        **if** $d_t^l = d_t^u = 0$ **then**
7           **while** $t \notin \mathcal{T}_L$ **do**
8              $t \leftarrow 2t + 1$
9           **end**
10          $\mathbb{T} \leftarrow \mathbb{T} \cup \{t\}$;
11       **else**
12          $F_t = \{j \mid (a_{jt}^l = a_{jt}^u = 1) \vee (a_{jt}^l \neq a_{jt}^u), \forall j \in \{1, \cdots, P\}\}$;
13          **for** *Each split feature* $f \in F_t$ **do**
14             **if** $\mathbf{x}_{sf} < b_t^l$ **then**
15                $\mathbb{T} \leftarrow \mathbb{T} \cup \{2t\}$;
16             **else if** $\mathbf{x}_{sf} \geq b_t^u$ **then**
17                $\mathbb{T} \leftarrow \mathbb{T} \cup \{2t + 1\}$;
18             **else**
19                $\mathbb{T} \leftarrow \mathbb{T} \cup \{2t, 2t + 1\}$ ;        /\* $\mathbf{x}_{sf} \in [b_t^l, b_t^u)$ \*/
20             **end**
21          **end**
22       **end**
23    **else**
24       $\mathcal{T}_{z_s} \leftarrow \mathcal{T}_{z_s} \cup \{t\}$;
25       **if** $y_{sk} \in [c_{kt}^l, c_{kt}^u], \forall k \in \{1, \cdots, K\}$ **then**
26          $\mathbb{L} \leftarrow \mathbb{L} \cup \{0\}$;
27       **else**
28          $\mathbb{L} \leftarrow \mathbb{L} \cup \{1\}$;
29       **end**
30    **end**
31 **end**
32 **return** $\mathcal{T}_{z_s}, L = \frac{1}{\hat{L}} \min\{l \mid l \in \mathbb{L}\} + \frac{\lambda}{n} \sum\limits_{t \in \mathcal{T}_D} d_t^l$;

---

### C.1 Proof of Theorem 2

In this part we prove Theorem 2 that Algorithm 2 can find all leafs that sample $s$ can fall into under $M$ and can obtain the global optimum for the lower bound subproblem $\beta_s(M)$.

*Proof.* We proof by induction on maximum depth $D$:

Base case: Suppose maximum depth $D = 0$, then there's only one leaf node $t = 1$ for the tree model, $\mathcal{T}_z = \{t\}$ under $M$. The statement is true.

Induction: Suppose the statement is true for the decision tree problem with maximum depth $D \geq 0$ under any specific $M$, then for decision tree problem with maximum depth $D + 1$, since $D \geq 0$, the first node is a decision node and line 6-22 guarantee to put all its possible descendent nodes into the node set $\mathbb{T}$ according to bound $M$ at decision node $t$. Since each possible descendent node can be regarded as a root node of decision tree model with depth $D$, then the decision tree problem rooted with the descendent nodes can find all possible leaf nodes that sample $s$ can fall into with given bound $M = [m^l, m^u]$, according to the assumption. Thus the statement is proved true on decision tree problem with maximum depth $D + 1$.

Therefore, since Algorithm 2 exhaust all possible loss of sample $s$ and pick the minimum one, Algorithm 2 will find the global optimum of subproblem $\beta_s(M) := \min_{m \in M} Q_s(m)$. $\qquad\square$

## D  Additional Analysis for Sample Reduction

In this section, we provide the new formula for two tighter lower bound strategies after sample reduction.

**Relaxed MIP**  With the assistance of $\mathcal{S}_{dt}$ we can further simplify Problem 7 as:

$$\beta^R(M) = \min_{c,L} \sum_{s \in \mathcal{S}_{ud}} \frac{1}{\hat{L}} L_s + \lambda \sum_{t \in \mathcal{T}_D} d_t^l + \sum_{s' \in \mathcal{S}_{dt}} \frac{1}{\hat{L}} L_{s'} \tag{19a}$$

$$\text{s.t. } L_s \geq 1 - \sum_{t \in \mathcal{T}_{z_s}} \sum_{k \in \mathcal{K}} y_{sk} c_{kt} \tag{19b}$$

$$\sum_{k \in \mathcal{K}} c_{kt} = 1, \quad \forall t \in \mathcal{T}_L \tag{19c}$$

$$0 \leq L_s \leq 1 \tag{19d}$$

$$c_{kt} \in \{0,1\}, \quad \forall t \in \mathcal{T}_L \tag{19e}$$

$$s \in \mathcal{S}_{ud} \tag{19f}$$

Here, $L_{s'}$ are determined according to sample reduction. Compared to Problem 7, the number of variable $L_s$ is reduced from $n$ to $|\mathcal{S}_{ud}|$. The solving efficiency of the second lower bound strategy can be further enhanced.

**Group decomposition**  The computing load of the subproblem formed by each group can also be reduced through sample reduction. Specifically, we can have the following equation for each subproblem $\beta_g^G$:

$$\beta_g^G(M) = \min_{m_g \in M, L_s} \sum_{s \in \mathcal{S}_{ud} \cap \mathcal{S}_g} \left( \frac{1}{\hat{L}} L_s + \frac{\lambda}{n} \sum_{t \in \mathcal{T}_D} d_t \right) + \sum_{s' \in \mathcal{S}_{dt} \cap \mathcal{S}_g} \frac{1}{\hat{L}} L_{s'} \tag{20a}$$

$$\text{s.t. Constraint } 2g, 2h, 2i, 2j, 2k \tag{20b}$$

$$\frac{1}{2} \sum_{k \in \mathcal{K}} (y_{sk} + c_{kt} - 2y_{sk} c_{kt}) - L_s \leq 1 - z_{st}, \; \forall t \in \mathcal{T}_{z_s} \tag{20c}$$

$$\sum_{k \in \mathcal{K}} c_{kt} = 1, \; \forall t \in \mathcal{T}_L \tag{20d}$$

$$\sum_{t \in \mathcal{T}_{z_s}} z_{st} = 1 \tag{20e}$$

$$\mathbf{a}_m^T (\mathbf{x}_s + \epsilon - \epsilon_{min}) + \epsilon_{min} \leq b_m + (1 + \epsilon_{max})(1 - z_{st}),$$
$$\forall m \in A_L(t), \; t \in \mathcal{T}_{z_s} \tag{20f}$$

$$\mathbf{a}_m^T \mathbf{x}_s \geq b_m - (1 - z_{st}), \quad \forall m \in A_R(t), \; t \in \mathcal{T}_{z_s} \tag{20g}$$

$$z_{st} \in \{0,1\}, \; \forall t \in \mathcal{T}_{z_s} \tag{20h}$$

$$c_{kt} \in \{0,1\}, \; \forall t \in \mathcal{T}_L \tag{20i}$$

$$s \in \mathcal{S}_{ud} \cap \mathcal{S}_g \tag{20j}$$

Here, we can see that the number of decision variables is also vastly reduced. The number of constraints related to samples is significantly reduced accordingly under the scheme of sample reduction. In the case that $|\mathcal{T}_{z_s}| = 1$ for some $s$, the corresponding constraint 20e - 20g are eliminated on these $s$, since here $z_s$ is fully determined.

# E    Additional Experiments

## E.1    Results with Hyperparameter Tuning

We implement the experiments with hyperparameter tuning and compared the performance of RS-OCT with CART, OCT [Bertsimas and Dunn, 2017], and FlowOCT [Aghaei et al., 2020]. Since FlowOCT is designed for datasets with binary features, all datasets are transformed to binary with one-hot encoding before feeding to FlowOCT. The code of FlowOCT is implemented in Python[†] and use Gurobi as the MIP solver. Table 3 and 4 present the results of training optimal classification trees on all datasets with hyperparameters selected through validation for all four methods. For OCT and our method, we performed hyperparameter tuning following the instruction in [Bertsimas and Dunn, 2017] to select the continuous regularization parameter $\lambda$. For CART, we first train a full decision tree with the maximum depth and then we prune the tree by tuning the pruning-purity parameter from the Julia package `DecisionTree.jl`. For FlowOCT, we followed the instruction in [Aghaei et al., 2020] and modify its regularization parameter $\lambda$ by selecting values in the set $\{0, 0.1, \cdots, 0.9\}$. From the result, we can see that even after hyperparameter tuning, RS-OCT still competitive on most of datasets, compared to other methods. Particularly, for large datasets (i.e., $n \geq 7,000$), when maximum depth of tree is two, RS-OCT can improve the training accuracy of 2.7% and 2.5% on average, compared to CART and OCT, respectively. The testing accuracy (out of sample accuracy) can be improved by 2.8% and 2.5% on average, compared to CART and OCT, respectively.

## E.2    Analysis and Ablation Test for Sample Reduction

We provide the analysis and ablation test of the sample reduction on small and median dataset to demonstrate its effectiveness. First, four indicators are presented to help evaluate the effect of sample reduction on each dataset. The NODES represents how many BB nodes are explored during the running of the BB algorithm. The MAX-LEVEL represents the maximum depth of the BB nodes explored by the BB algorithm. The MAX-RATE and AVG-RATE are the indicators of the maximum and average percent of samples whose loss are pre-determined before the calculation of each node. We did not put the MIN-RATE here since, in the root decision node of the tree, no sample can be determined, and thus the MIN-RATE is always zero.

The numerical results (Table 5) reveal that the effect of sample reduction will primarily reduce the computation load when the search tree of the BB algorithm goes deep. Specifically, for datasets (SEEDS, GLASS, BODY, BANKNOTE, WALL-FOLLOWING) that can converge to a small optimality gap (e.g.$\leq 1$), the sample reduction devote a significant effort on the calculation process in terms of Max-rate and Avg-Rate. Almost all of these datasets can, on maximum, have the loss determined for over $90\%$ of samples and maintain a relatively high average rate of determined samples among all explored nodes. For the indicator of Nodes and Max-Level, we discover that the exploration of more nodes could also harness the effect of sample reduction even if the node's level is not deep, especially when the data size is large.

Next, we perform the ablation test for sample reduction. To make the comparison more precise, we only test on datasets where the average percentage of determined samples is above zero. Table 6 presents the test result and reveals that, with sample reduction, all datasets can either converge to a smaller optimality gap within the same time or use less time to reach <1% optimality gap. Therefore, the sample reduction can have a recognizable effect on improving the calculation efficiency of each BB node.

One limitation of the sample reduction is that it could have reduced influence on datasets with high feature dimensions (e.g. Both two OZONE datasets have 72 features). The reason could be probably due to the large search space of the decision variables (i.e. The number of first-stage variable is $(P+2)|\mathcal{T}_D| + K|\mathcal{T}_L|$, which is proportional to $P$ and $K$). Even if the bounds of decision variables is narrowed through the BB procedure, the feasible space under a BB node is still too large to help determine the loss of any samples.

---

[†]https://github.com/pashew94/StrongTree

Table 3: Numerical results on small and median datasets with hyperparameter tuning (Serial).

| DATA-SET | $n$ | $P$ | $K$ | METHOD | DEPTH = 2 | | DEPTH = 3 | |
|---|---|---|---|---|---|---|---|---|
| | | | | | TRAINING ACCURACY (%) | TESTING ACCURACY (%) | TRAINING ACCURACY (%) | TESTING ACCURACY (%) |
| SEEDS | 210 | 7 | 3 | CART | 91.2 | **94.1** | 91.8 | **94.1** |
| | | | | OCT | **94.3** | 86.3 | **96.9** | 90.2 |
| | | | | FlowOCT | 43.8 | 30.8 | 45.3 | 32.5 |
| | | | | RS-OCT | **94.3** | 88.2 | 95.0 | **94.1** |
| GLASS | 214 | 9 | 6 | CART | 65.8 | **64.3** | 72.2 | 64.3 |
| | | | | OCT | 67.2 | **64.3** | **72.8** | **67.9** |
| | | | | FlowOCT | 55.1 | 40.7 | 59.2 | 37.0 |
| | | | | RS-OCT | **68.4** | 62.5 | 71.5 | 62.5 |
| BODY | 507 | 5 | 2 | CART | 90.0 | 88.9 | 91.9 | 88.1 |
| | | | | OCT | 91.1 | 87.3 | **93.7** | **93.7** |
| | | | | FlowOCT | 60.6 | 56.7 | 63.9 | 59.8 |
| | | | | RS-OCT | **92.1** | **92.9** | **93.7** | 91.3 |
| STATLOG-GERMAN | 1,000 | 24 | 2 | CART | 73.7 | 71.2 | 76.7 | **74.8** |
| | | | | OCT | 73.9 | 70.0 | 75.5 | 72.4 |
| | | | | FlowOCT | **75.7** | **72.0** | **77.2** | 72.0 |
| | | | | RS-OCT | 73.9 | 70.0 | 76.0 | 70.8 |
| CONCRETE | 1,030 | 8 | 3 | CART | 62.6 | 61.9 | 69.5 | 65.0 |
| | | | | OCT | **65.3** | 63.0 | **69.9** | 64.6 |
| | | | | FlowOCT | 57.6 | 61.2 | 66.3 | 65.1 |
| | | | | RS-OCT | 65.2 | **63.8** | 67.8 | **70.4** |
| BANKNOTE | 1,372 | 4 | 2 | CART | 90.0 | 89.0 | 95.1 | 95.9 |
| | | | | OCT | 90.3 | 87.5 | 96.2 | **98.5** |
| | | | | FlowOCT | 54.0 | 52.2 | 55.1 | 52.2 |
| | | | | RS-OCT | **92.7** | **91.3** | **97.9** | 96.8 |
| CONTRA-CEPTIVE | 1,473 | 11 | 3 | CART | 47.7 | 46.3 | 53.4 | 52.6 |
| | | | | OCT | **52.0** | **52.6** | **55.8** | 56.4 |
| | | | | FlowOCT | 50.3 | 44.4 | 53.4 | 46.3 |
| | | | | RS-OCT | **52.0** | **52.6** | 55.6 | **56.6** |
| OZONE-EIGHT | 1,847 | 72 | 2 | CART | 93.1 | 93.1 | **93.7** | 92.0 |
| | | | | OCT | 93.6 | 93.3 | **93.7** | **92.8** |
| | | | | FlowOCT | 93.2 | **93.7** | 7.4 | 6.5 |
| | | | | RS-OCT | **93.8** | 92.6 | **93.7** | **92.8** |
| OZONE-ONE | 1,848 | 72 | 2 | CART | 97.0 | **96.8** | 97.0 | 96.8 |
| | | | | OCT | 97.1 | 96.3 | **97.3** | 96.3 |
| | | | | FlowOCT | 96.8 | 95.7 | 96.3 | **97.2** |
| | | | | RS-OCT | **97.3** | **96.8** | 97.1 | 96.1 |
| THYROID-ANN | 3,772 | 21 | 3 | CART | **97.8** | **98.1** | **99.4** | 99.0 |
| | | | | OCT | **97.8** | **98.1** | **99.4** | 98.8 |
| | | | | FlowOCT | 92.9 | 94.4 | 93.0 | 94.4 |
| | | | | RS-OCT | **97.8** | **98.1** | **99.4** | **99.2** |
| STATLOG-LANSAT | 4,435 | 36 | 6 | CART | 64.7 | 64.6 | 79.6 | 79.8 |
| | | | | OCT | 65.2 | 64.8 | 79.6 | 79.2 |
| | | | | FlowOCT | 31.6 | 33.0 | 33.8 | 33.3 |
| | | | | RS-OCT | **66.6** | **67.8** | **80.1** | **79.9** |
| SPAMBASE | 4,601 | 57 | 2 | CART | **86.6** | **86.2** | 87.9 | 87.0 |
| | | | | OCT | 86.0 | 84.5 | 88.6 | 87.2 |
| | | | | FlowOCT | 81.3 | 82.8 | OOM[1] | OOM |
| | | | | RS-OCT | 85.4 | 84.5 | **89.6** | **89.0** |
| WALL-FOLLOWING | 5,456 | 2 | 4 | CART | **94.0** | **94.0** | 100.0 | 100.0 |
| | | | | OCT | **94.0** | **94.0** | 100.0 | 100.0 |
| | | | | FlowOCT | 42.5 | 40.9 | 43.1 | 41.4 |
| | | | | RS-OCT | **94.0** | **94.0** | 100.0 | 100.0 |
| PAGE-BLOCK | 5,473 | 10 | 5 | CART | 95.4 | 95.0 | 96.3 | 95.6 |
| | | | | OCT | **95.4** | **95.3** | **96.4** | **95.7** |
| | | | | FlowOCT | 93.9 | 92.3 | OOM | OOM |
| | | | | RS-OCT | **95.4** | **95.3** | **96.4** | **95.7** |

[1] OUT OF MEMORY.

Table 4: Numerical results on large datasets with hyperparameter tuning (Paralleled with 1000 cores).

| DATA-SET | $n$ | $P$ | $K$ | METHOD | DEPTH = 2 | | DEPTH = 3 | |
|---|---|---|---|---|---|---|---|---|
| | | | | | TRAINING ACCURACY (%) | TESTING ACCURACY (%) | TRAINING ACCURACY (%) | TESTING ACCURACY (%) |
| PENDIGITS | 7,494 | 16 | 10 | CART | 38.9 | 39.1 | 62.7 | 62.9 |
| | | | | OCT | 39.2 | 39.4 | 62.8 | 63.3 |
| | | | | FLOWOCT | 36.5 | 39.0 | 52.1 | 53.7 |
| | | | | RS-OCT | **39.9** | **39.9** | **63.9** | **63.7** |
| AVILA | 10,430 | 10 | 12 | CART | 52.2 | 52.9 | **55.0** | **55.3** |
| | | | | OCT | 52.4 | 53.3 | **55.0** | **55.3** |
| | | | | FLOWOCT | OOM[1] | OOM | OOM | OOM |
| | | | | RS-OCT | **53.5** | **54.6** | **55.0** | **55.3** |
| EEG | 14,980 | 14 | 2 | CART | 62.3 | 61.7 | 65.6 | 63.8 |
| | | | | OCT | 62.3 | 61.7 | 55.1 | 55.1 |
| | | | | FLOWOCT | 55.4 | 56.3 | 44.7 | 43.7 |
| | | | | RS-OCT | **66.7** | **66.0** | **67.9** | **65.7** |
| HTRU | 17,898 | 8 | 2 | CART | 97.8 | 97.7 | 97.8 | **97.7** |
| | | | | OCT | 97.8 | 97.7 | 97.8 | **97.7** |
| | | | | FLOWOCT | OOM | OOM | OOM | OOM |
| | | | | RS-OCT | **98.0** | **97.8** | **97.9** | **97.7** |
| SHUTTLE | 43,500 | 9 | 7 | CART | 93.9 | 93.8 | 99.6 | 99.6 |
| | | | | OCT | 93.9 | 93.8 | 99.6 | 99.6 |
| | | | | FLOWOCT | OOM | OOM | OOM | OOM |
| | | | | RS-OCT | **95.4** | **95.5** | **99.7** | **99.8** |
| SKIN-SEGMEN-TATION | 245,057 | 3 | 2 | CART | 90.7 | 90.6 | 93.8 | 93.8 |
| | | | | OCT | 90.7 | 90.6 | 93.8 | 93.8 |
| | | | | FLOWOCT | OOM | OOM | OOM | OOM |
| | | | | RS-OCT | **92.7** | **92.7** | **96.6** | **96.5** |
| HT-SENSOR | 928,991 | 11 | 3 | CART | 58.1 | 58.1 | 64.3 | 64.3 |
| | | | | OCT | 58.1 | 58.1 | 64.3 | 64.3 |
| | | | | FLOWOCT | OOM | OOM | OOM | OOM |
| | | | | RS-OCT | **59.7** | **59.8** | **64.6** | **64.5** |

[1] OUT OF MEMORY.

Table 5: The effects of sample reduction on small and median datasets ($D = 2, \lambda = 0.05$, serial).

| DATASET | $n$ | $P$ | $K$ | NODES | MAX-LEVEL | MAX-RATE (%) | AVG-RATE (%) |
|---|---|---|---|---|---|---|---|
| SEEDS | 210 | 7 | 3 | 345 | 38 | 96.9 | 35.6 |
| GLASS | 214 | 9 | 6 | 178 | 50 | 93.7 | 23.5 |
| BODY | 507 | 5 | 2 | 3195 | 43 | 100 | 60.5 |
| STATLOG-GERMAN | 1,000 | 24 | 2 | 3736 | 42 | 0.0 | 0.0 |
| CONCRETE | 1,030 | 8 | 3 | 22 | 10 | 0.0 | 0.0 |
| BANKNOTE | 1,372 | 4 | 2 | 9538 | 38 | 99.9 | 89.9 |
| CONTRACEPTIVE | 1,473 | 11 | 3 | 7553 | 30 | 84.3 | 3.9 |
| OZONE-EIGHT | 1,847 | 72 | 2 | 2012 | 73 | 0.0 | 0.0 |
| OZONE-ONE | 1,848 | 72 | 2 | 6181 | 83 | 0.0 | 0.0 |
| THYROID-ANN | 3,772 | 21 | 3 | 23142 | 79 | 97.8 | 0.8 |
| STATLOG-LANSAT | 4,435 | 36 | 6 | 1784 | 45 | 0.5 | 0.0 |
| SPAMBASE | 4,601 | 57 | 2 | 146 | 72 | 0.0 | 0.0 |
| WALL-FOLLOWING | 5,456 | 2 | 4 | 93 | 38 | 91.4 | 19.8 |
| PAGE-BLOCK | 5,473 | 10 | 5 | 4227 | 52 | 99.0 | 2.5 |

Table 6: Ablation test of sample reduction on small and median datasets ($D = 2, \lambda = 0.05$, serial).

| DATASET | $n$ | $P$ | $K$ | TIME-NO SR | TIME-SR | GAP-NO SR | GAP-SR |
|---|---|---|---|---|---|---|---|
| SEEDS | 210 | 7 | 3 | 329.1 | **151.2** | <1% | <1% |
| GLASS | 214 | 9 | 6 | 2,125.6 | **1,903.8** | <1% | <1% |
| BODY | 507 | 5 | 2 | 1,036.1 | **965.9** | <1% | <1% |
| BANKNOTE | 1372 | 4 | 2 | 773.1 | **633.8** | <1% | <1% |
| WALL-FOLLOWING | 5456 | 2 | 4 | 338.3 | **223.2** | <1% | <1% |
| CONTRACEPTIVE | 1473 | 11 | 3 | 14,400.0 | 14,400.0 | 12.2% | **12.0%** |
| THYROID-ANN | 3772 | 21 | 3 | 14,400.0 | 14,400.0 | 68.7% | **55.6%** |
| PAGE-BLOCK | 5473 | 10 | 5 | 14,400.0 | 14,400.0 | 47.3% | **46.2%** |