# OpenReview forum: "A Scalable Deterministic Global Optimization Algorithm for Training Optimal Decision Tree"
_NeurIPS.cc/2022/Conference — NeurIPS 2022 Accept_

### Official Review · Reviewer_v4RW · 2022-07-11

**Rating:** 4
**Confidence:** 3
**Soundness:** 2 fair
**Presentation:** 2 fair
**Contribution:** 3 good

**Summary:**

This paper presents an algorithm for learning the optimal decision tree w.r.t. regularized training loss. The algorithm is based on the branch-and-bound method, and takes a two-stage stochastic optimization approach to decompose the problem. The main novelty of the algorithm is the computation of the dual bound at each search node, where the authors: use separability to split the second stage problem into many "group" problems solved in parallel, and introduce a data reduction scheme to further reduce the node problem sizes. The authors present a computational study comparing their method against existing methods for training tree models in terms of model accuracy, optimality gap, and solve time.

**Questions:**

* The typesetting of (2) appears pretty broken: Missing "for all" quantifiers, (2m) does not seem to make sense as s is not a decision variable, etc.
* Please situate (2) in the literature: Is this equivalent to the formulation of Bertsimas and Dunn, or are their modifications for which you are claiming novelty? If so, please highlight those modifications and, importantly, provide a formal proof of the validity of the formulation in the appendix.
* "BB node" is used in the abstract and introduction, but is only defined at the end of the introduction.
* The alpha symbol is reused with different meanings in (1) and Section 3.
* I believe you want to flip the min and sum in (6) and (8).

**Limitations:**

The authors do not explicitly address the potential negative social impact of their work. The discussion of limitations is terse, but sufficient.

**Strengths And Weaknesses:**

The paper is situated in an area of interest and significance to the NeurIPS community, and presents algorithmic improvements that are, to my knowledge, original. The computational results suggest an improvement over the state of the art, in terms of model accuracy and scalability of the training algorithm. The paper is somewhat difficult to read at points, and does not do as much to contextualize the results with the existing literature as it could.

My main concern with the paper is the hyperparameter selection in the computational study. The authors select a single regularization parameter value alpha for the entire study, seemingly arbitrarily. This regularization parameter can have a substantial impact on the model accuracy, as well as potentially on the solve time. It is not obvious to me that choosing the same value for the different algorithms is the "right" thing to do in order to get the highest quality models. It is certainly not the case that this value should remain fixed for different data sets, as the authors acknowledge on page 9.

Therefore, I think the computational study would be much stronger the regularization hyperparameter is selected through validation, done separately for each of the three methods and each of the data sets.

---

> ### Author Response · Authors · 2022-08-02
> **Response to Reviewer v4RW**
>
> We sincerely thank the reviewer for their time and thoughtful comments. Here is our response to the reviewer's comments.
>
> * **Q1 - My main concern with the paper is the hyperparameter selection in the computational study...Therefore, I think the computational study would be much stronger the regularization hyperparameter is selected through validation, done separately for each of the three methods and each of the data sets.**
>
>  Thank you very much for pointing out this issue. This paper focuses on improving the solution process of the optimal decision tree problem. Therefore, in our original submission, we use the same value of $\alpha$, so that both our method and OCT are solving the same optimization problem, and we can have a fair comparison of the optimality gap, solution time, and best achieved upper bound on this specific problem formulation.
>
>  After reading the reviewer's comments, we totally agree that, to have a fair comparison on the test accuracy, it is the *right* thing to do the hyperparameter tuning. Following the reviewer's suggestion, we performed additional experiments, and the following table presents the new results of training optimal classification trees on all small and median-sized datasets with hyperparameters selected through validation for all three methods. For OCT and our method, we performed hyperparameter tuning following the instruction in [1] to select the continuous regularization parameter $\alpha$. For CART, we first train a full decision tree with the maximum depth and then we prune the tree by tuning the pruning_purity parameter from the Julia package `DecisionTree.jl`.
>
>
>  | Dataset | n | P | K | Train CART | Train OCT | Train RSOCT no-val | Train RSOCT | Test CART | Test OCT | Test RSOCT no-val | Test RSOCT |
>  |----------------|------|----|---|------------|-----------|--------------|-----------|-----------|----------|-------------|----------|
>  | Seeds | 210 | 7 | 3 | 91.2 | **94.3** | **94.3** | **94.3** | **94.1** | 86.3 | 88.2 | 88.2 |
>  | Glass | 214 | 9 | 6 | 65.8 | 67.2 | **68.4** | **68.4** | **64.3** | **64.3** | 62.5 | 62.5 |
>  | Body | 507 | 5 | 2 | 90.0 | 91.1 | **92.1** | **92.1** | 88.9 | 87.3 | **92.9** | **92.9** |
>  | Statlog-German | 1000 | 24 | 2 | 73.7 | 73.9 | **75.1** | 73.9 | 71.2 | 70.0 | **72.0** | 70.0 |
>  | Concrete | 1030 | 8 | 3 | 62.6 | **65.3** | 65.2 | 65.2 | 61.9 | 63.0 | 63.0 | **63.8** |
>  | Banknote | 1372 | 4 | 2 | 90.0 | 90.3 | **92.7** | **92.7** | 89.0 | 87.5 | **92.2** | 91.3 |
>  | Contraceptive | 1473 | 11 | 3 | 47.7 | 52.0 | **54.0** | 52.0 | 46.3 | 52.6 | **56.4** | 52.6 |
>  | Ozone-eight | 1847 | 72 | 2 | 93.1 | 93.6 | 93.7 | **93.8** | 93.1 | **93.3** | 92.4 | 92.6 |
>  | Ozone-one | 1848 | 72 | 2 | 97.0 | 97.1 | 97.2 | **97.3** | **96.8** | 96.3 | **96.8** | **96.8** |
>  | Thyroid-ann | 3772 | 21 | 3 | **97.8** | **97.8** | **97.8** | **97.8** | **98.1** | **98.1** | **98.1** | **98.1** |
>  | Statlog-lansat | 4435 | 36 | 6 | 64.7 | 65.2 | 65.4 | **66.6** | 64.6 | 64.8 | 66.0 | **67.8** |
>  | Spambase | 4601 | 57 | 2 | 86.6 | 86.0 | **86.7** | 85.4 | **86.2** | 84.5 | 85.9 | 84.5 |
>  | Wall-Following | 5456 | 2 | 4 | **94.0** | **94.0** | **94.0** | **94.0** | **94.0** | **94.0** | **94.0** | **94.0** |
>  | Page-block | 5473 | 10 | 5 | **95.4** | **95.4** | **95.4** | **95.4** | 95.0 | **95.3** | 95.0 | **95.3** |
>
>  *Here, "OCT" represents the work from [1], and "no_val" represents the result of our method without hyperparameter tuning. All other results without "no_val" are obtained after hyperparameter tuning.*
>
>  From the table, we can see that with the hyperparameter tuning, our method still performs quite well in most datasets, especially for datasets with over 1000 samples.
>
>  Due to the limitation of our computation resources, we currently only test results for small and median-sized datasets. We will add more results on the large datasets in the final version.

---

> > ### Author Response · Authors · 2022-08-02
> > **Response to Reviewer v4RW**
> >
> > * **Q2 - Other concerns.**
> >  * **The typesetting of (2) appears pretty broken: Missing "for all" quantifiers, (2m) does not seem to make sense as s is not a decision variable, etc.**
> >  * **Please situate (2) in the literature: Is this equivalent to the formulation of Bertsimas and Dunn, or are their modifications for which you are claiming novelty? If so, please highlight those modifications and, importantly, provide a formal proof of the validity of the formulation in the appendix.**
> >  * **"BB node" is used in the abstract and Introduction, but is only defined at the end of the Introduction.**
> >  * **The alpha symbol is reused with different meanings in (1) and Section 3.**
> >  * **I believe you want to flip the min and sum in (6) and (8).**
> >
> >
> >  We apologize for the confusing notations and typos. Your suggestions will be incorporated into the final version of the paper. We will fix the typos, rewrite the formulation (2), and clarify all the notations as the reviewer suggested.
> >
> > We truly appreciate your careful and detailed review and apologize for the typo made in our paper. We will follow your suggestions and make corresponding modifications in the revised article.
> >
> > [1] Bertsimas, D., & Dunn, J. (2017). Optimal classification trees. Machine Learning, 106(7), 1039-1082. ISO 690

---

> > > ### Author Response · Authors · 2022-08-08
> > > **Additional Response to Reviewer v4RW**
> > >
> > > We performed additional experiments (we just exhausted our HPC resources:)) and hope the new results can resolve the reviewer's concerns. We are glad to receive the valuable comments, and it is always welcome to ask more questions about our response. We hope that through the rebuttal and discussion, we will convince the reviewer to raise their evaluation. We would really appreciate the support!
> > >
> > > * **Experiments on Large Dataset with Hyperparameter Tuning**
> > >
> > > Here, we provide the results on large datasets (n > 200,000) with Hyperparameter Tuning.
> > >
> > > Table for Training Optimal Classification Tree with Maximum Depth=2 on Large Datasets with 1000 Cores
> > >
> > > | Dataset | n | P | K | Train-CART | Train-OCT | Train-no_val | Train-RSOCT | Test-CART | Test-OCT | Test-no_val | Test-RSOCT |
> > > |-------------------|--------|----|---|------------|-----------|--------------|-----------|-----------|----------|-------------|----------|
> > > | Skin-Segmentation | 245057 | 3 | 2 | 90.7 | 90.7 | **92.7** | **92.7** | 90.6 | 90.6 | **92.7** | **92.7** |
> > > | HTS | 928991 | 11 | 3 | 58.1 | 58.1 | **59.7** | **59.7** | 58.1 | 58.1 | **59.7** | **59.8** |
> > >
> > >  *"no_val" represents the result of our method without hyperparameter tuning. All other results without "no_val" are obtained after hyperparameter tuning.*
> > >
> > > This table shows that our algorithm (RSOCT) outperforms CART and OCT in both the training and testing after hyperparameter tuning, even for large datasets.
> > >
> > > * **Significance.**
> > >
> > > We would like to reiterate that we are the first to provide such a framework for training the optimal decision tree on a large dataset (continuous features with up to almost **1 million** samples) with the following attributes:
> > >
> > > 1) We reconstruct the optimal decision tree training as a two-stage stochastic programming problem and provide a reduced-space branch and bound algorithm for this problem.
> > > 2) We prove that our algorithm can guarantee convergence by only branching on the variables that determine the tree structure (i.e. variable $a,b,c,d$ in Formula (2)).
> > > 3) Our new MIP formulation consists of a decomposable structure that can be directly parallelized when solving the lower bounding problem. This attribute enables our algorithm to scale to large datasets through trivial parallelism.
> > > 4) We also proposed the sample reduction method to predetermine the cost of each sample before calculating each node's lower bound.

---

### Official Review · Reviewer_gshW · 2022-07-11

**Rating:** 7
**Confidence:** 4
**Soundness:** 3 good
**Presentation:** 3 good
**Contribution:** 3 good

**Summary:**

The authors propose an approach for finding exactly optimal decision trees using a two stage stochastic program, with the first stage problem consisting of the variables related to the structure of the tree such as splits, and the second stage variables consisting of example-related terms such as what leaves the sample falls into as it progresses in the tree structure. The approach uses a modified branch and bound approach to solve this two-stage problem and demonstrates improved runtime performance and testing accuracy over baselines in larger datasets which are either heuristic decision tree methods or optimal decision trees which are formulated as large mixed integer programs.

**Questions:**

Can individual components of the proposed approach be integrated into previous work on optimal classification trees?

Does the proposed approach close the gap faster in terms of runtime? What does the gap convergence here look like compared to previous approaches? It would be helpful to give an idea of what this looks like throughout the solving process to understand how beneficial/harmful it would be to stop the solving procedure early given time constraints.

Is it possible to compute the gap for the CART models? Here you still have the quality of the primal solution and you might use one of the computed bounds to determine how suboptimal the CART solution is.

Is it possible to encode rules enforcing fairness into the solving procedure? Is the formulation flexible enough to address other objectives in the training procedure?

Small changes:
Line 50: handle large dataset[s]


**Limitations:**

The authors adequately address the limitation of only being able to solve for trees with limited depth as previous work is also limited by this constraint. It is a promising direction for future work but scalability here is one main limitation of the current work in optimal decision trees.

**Strengths And Weaknesses:**

Overall the paper’s main strength is making the guarantees of optimal decision trees applicable for larger datasets by proposing an intricate solving approach that enables efficient closing of the primal and dual bound. Furthermore, it Is able to demonstrate the benefit of having optimal classification trees by finding improved solutions that outperform standard methods in terms of test time performance. The proposed approach not only gives improved primal solutions in the case of large datasets, but also gives performance bounds for the dataset it is trained on and is able to determine how suboptimal the current solution can possibly be.

The paper itself is clear and well-written, adequately explaining the approaches in previous work and the significance with respect to the previous work. Furthermore, the solution algorithm is original and the different components seem to be well-founded in addressing limitations of previous work.

The proposed approach has several components that tackle different challenges of the optimal decision tree problem. It would strengthen the paper to display results corresponding to the strengths of the work to validate the approach. Specifically, it would be interesting to see the gap convergence plots for the given approach and potentially a baseline to show that the work done for improving the bound does indeed result in tighter bounds and better gap closure. Additionally, it would be interesting to see if these types of approaches could be integrated into previous work to understand the importance of the individual contributions.

---

> ### Author Response · Authors · 2022-08-02
> **Response to Reviewer gshW**
>
> We appreciate the reviewer's insightful and impressive comments. Here is our response to your concerns as follows.
>
> * **Can individual components of the proposed approach be integrated into previous work on optimal classification trees?**
>
>  Yes, as long as the proposed formula can be written as a form of a stochastic programming problem, all of our proposed components, including the reduced-space branch and bound, lower and upper bounding methods, and sample reduction methods, can be integrated for training the optimal classification tree. For example, the proposed formulation of Verwer and Zhang [1] (works on binary features) and Zhu et al. [2] (works on the optimal multivariate trees) can potentially be reformulated as a stochastic programming problem, and it would be straightforward to tailor our approach for these formulations.
>
>  We will add this as an interesting direction for future work.
>
> * **Does the proposed approach close the gap faster in terms of runtime? What does the gap convergence here look like compared to previous approaches? It would be helpful to give an idea of what this looks like throughout the solving process to understand how beneficial/harmful it would be to stop the solving procedure early given time constraints.**
>
>  Thanks for highlighting this issue. Here we provide an example of the gap convergence graph for the *Banknote dataset*. We will add all gap convergence plots in the appendix of the revised version. (Since OpenReview can not embed links with pictures, we put the graph into an anonymous GitHub repository for your reference: https://github.com/RS-OCT/rebuttal-picture/blob/main/Banknote-authentication-sd1-2.png. We apologize for the inconvenience. )
>
>  From the graph, we can see that the optimal solution (the best upper bound) could be obtained in the first several iterations, and the rest of the iterations are just spent on verifying the optimality. In practice, users could stop the solving procedure early to get a near-optimal solution if the proof of optimality is not necessary.
>
>  In contrast, OCT (the previous approach we compare with) still has a large optimality gap (over 98%) after 4 hours of running for most datasets. For most large datasets, the UB provided by OCT after 4 hours is the same as CART, and is much worse than our approach.
>
> * **Is it possible to compute the gap for the CART models? Here you still have the quality of the primal solution and you might use one of the computed bounds to determine how suboptimal the CART solution is.**
>
>  We appreciate the reviewer's suggestions. Although CART itself can not provide a gap to quantify the quality of its solution, we can also use our method's final calculated lower bound to evaluate the suboptimality of the CART solution. In this aspect, our method provides a baseline to measure the goodness of the solution provided by any heuristic decision tree algorithm on a particular dataset.
>
> * **Is it possible to encode rules enforcing fairness into the solving procedure? Is the formulation flexible enough to address other objectives in the training procedure?**
>
>  Thank you for reminding us about the fairness issue. One of the most popular formulations for learning fair decision trees is proposed by Aghaei et al.[3], which penalizes the unfairness of using a regularizer in the objective function. This formulation can be reformulated as a stochastic optimization problem, and we can tailor our approach to address this formulation.
>
> [1] Verwer, S., & Zhang, Y. (2019, July). Learning optimal classification trees using a binary linear program formulation. In Proceedings of the AAAI conference on artificial intelligence (Vol. 33, No. 01, pp. 1625-1632).
>
>  [2] Zhu, H., Murali, P., Phan, D., Nguyen, L., & Kalagnanam, J. (2020). A scalable mip-based method for learning optimal multivariate decision trees. Advances in Neural Information Processing Systems, 33, 1771-1781.
>
>  [3] Aghaei, S., Azizi, M. J., & Vayanos, P. (2019, July). Learning optimal and fair decision trees for non-discriminative decision-making. In Proceedings of the AAAI Conference on Artificial Intelligence (Vol. 33, No. 01, pp. 1418-1426).
>
> Again we appreciate the reviewer's constructive comments, and all suggestions will be incorporated into the revised version of our work.

---

### Official Review · Reviewer_fTBj · 2022-07-11

**Rating:** 6
**Confidence:** 4
**Soundness:** 3 good
**Presentation:** 3 good
**Contribution:** 3 good

**Summary:**

The paper presents a new approach for globally optimal decision trees. Inspired by the scalability issues of previous work, the paper proposes to formulate the problem as a two-stage optimization problem and develop a specialized branch and bound approach for training optimal decision trees for classification problems. The approach uses decomposition of bound computation that can be parallelized as well as a sample reduction technique. Experiments show the approach outperforms the well-known OCT approach and the heuristic algorithm CART.


-----
I thank the authors for their thoughtful response and their attempt to address the comments. Although the new results do not cover most of the approaches I suggested, I acknowledge it may be challenging to obtain and run the code in short time. It would be useful to add at least some of them once you successfully run them.

Also, the results for depth of 3 are promising and show the proposed approach outperforms the baselines.

I have therefore increased my score.

**Questions:**

I don't have any questions beyond my points above

**Limitations:**

I did not identify any issues related to limitations and potential negative societal impact.

**Strengths And Weaknesses:**

Strengths:
- Interesting and novel approach for globally optimal decision trees based on two-stage optimization and specialized branch and bound algorithm.
- Strong theoretical guarantees and focus on both quality solution and the optimality gap.
- Experiments show the approach significantly outperforms the baselines.

Weaknesses:
- Missing important baselines in experiments: current experiments only compare to OCT and ignore a large amount of approaches for optimal decision trees developed over the last year. In particular, despite the focus on MIP and continuous features, the work should compare against prominent non-MIP-based optimal approaches and approaches that first convert continuous values to discrete values using a transformation that maintains optimality (e.g., assigning a single variable to each value). Some examples include [1-5] listed below.
- Experiments are limited to trees of depth 2. This is a very shallow depth and many previous works have considered deeper trees.
- Ablation study: the paper would benefit from an analysis of the impact of different components (e.g., performance without the sample reduction technique).
- Minor points:
	* line 3, 36: "tailed" I think should be "tailored"?
	* line 104: "feature size" -> "number of features"?
	* line 214: "reduce" -> "reduces"

[1] Verwer, Sicco, and Yingqian Zhang. "Learning optimal classification trees using a binary linear program formulation." Proceedings of the AAAI conference on artificial intelligence. Vol. 33. No. 01. 2019.
[2] Aghaei, Sina, Andrés Gómez, and Phebe Vayanos. "Strong optimal classification trees." arXiv preprint arXiv:2103.15965 (2021).
[3] Aglin, Gaël, Siegfried Nijssen, and Pierre Schaus. "Learning optimal decision trees using caching branch-and-bound search." Proceedings of the AAAI Conference on Artificial Intelligence. Vol. 34. No. 04. 2020.
[4] Hu, Hao, et al. "Learning optimal decision trees with maxsat and its integration in adaboost." IJCAI-PRICAI 2020, 29th International Joint Conference on Artificial Intelligence and the 17th Pacific Rim International Conference on Artificial Intelligence. 2020.
[5] Shati, Pouya, Eldan Cohen, and Sheila McIlraith. "SAT-based approach for learning optimal decision trees with non-binary features." 27th International Conference on Principles and Practice of Constraint Programming (CP 2021). Schloss Dagstuhl-Leibniz-Zentrum für Informatik, 2021.

---

> ### Author Response · Authors · 2022-08-02
> **Response to Reviewer fTBj**
>
> We appreciate the reviewer's careful and insightful comments. Due to the limited computation resources available to us, we currently only tested results for small and median-sized datasets. We will add more results in the final version of the paper. Here are the detailed replies to your concerns.
>
> * **Missing important baselines in experiments: current experiments only compare to OCT and ignore a large amount of approaches for optimal decision trees developed over the last year. In particular, despite the focus on MIP and continuous features, the work should compare against prominent non-MIP-based optimal approaches and approaches that first convert continuous values to discrete values using a transformation that maintains optimality (e.g., assigning a single variable to each value). Some examples include [1-5] listed below.**
>
>
> Thank you for suggesting alternative methods of comparison. We will add a discussion and more numerical experiments to compare their performance.
>
> As the reviewer suggested, we tried to execute numerical experiments using methods in [1-5]. However, we have trouble running the codes provided in these papers. We guess it is because some libraries used in these papers have already been updated (We have contacted the authors but have not received any reply yet). We currently can only run the flowOCT method [2]. We transform the continuous feature into binary by assigning a single variable to each possible value. The following table presents our renewed results with the flowOCT method [2] after hyperparameter tuning. This table shows that the flowOCT method [2] does not work well for continuous datasets.
>
>
>  Table for Training Optimal Classification Tree with maximum depth=2
>
>  | Dataset | n | P | K | Train-CART | Train-OCT | Train-Flow | Train-RSOCT | Test-CART | Test-OCT | Test-Flow | Test-RSOCT |
>  |----------------|------|----|---|------------|-----------|------------|-------------|-----------|----------|-----------|------------|
>  | Seeds | 210 | 7 | 3 | 91.2 | **94.3** | 43.8 | **94.3** | **94.1** | 86.3 | 30.8 | 88.2 |
>  | Glass | 214 | 9 | 6 | 65.8 | 67.2 | 55.1 | **68.4** | **64.3** | **64.3** | 40.7 | 62.5 |
>  | Body | 507 | 5 | 2 | 90.0 | 91.1 | 60.6 | **92.1** | 88.9 | 87.3 | 56.7 | **92.9** |
>  | Statlog-German | 1000 | 24 | 2 | 73.7 | 73.9 | **75.7** | 73.9 | 71.2 | 70.0 | **72.0** | 70.0 |
>  | Concrete | 1030 | 8 | 3 | 62.6 | **65.3** | 57.6 | 65.2 | 61.9 | 63.0 | 61.2 | **63.8** |
>  | Banknote | 1372 | 4 | 2 | 90.0 | 90.3 | 54.0 | **92.7** | 89.0 | 87.5 | 52.2 | **91.3** |
>  | Contraceptive | 1473 | 11 | 3 | 47.7 | **52.0** | 50.3 | **52.0** | 46.3 | **52.6** | 44.4 | **52.6** |
>  | Ozone-eight | 1847 | 72 | 2 | 93.1 | 93.6 | 93.2 | **93.8** | 93.1 | 93.3 | **93.7** | 92.6 |
>  | Ozone-one | 1848 | 72 | 2 | 97.0 | 97.1 | 96.8 | **97.3** | **96.8** | 96.3 | 95.7 | **96.8** |
>  | Thyroid-ann | 3772 | 21 | 3 | **97.8** | **97.8** | 92.9 | **97.8** | **98.1** | **98.1** | 94.4 | **98.1** |
>  | Statlog-lansat | 4435 | 36 | 6 | 64.7 | 65.2 | 31.6 | **66.6** | 64.6 | 64.8 | 33.0 | **67.8** |
>  | Spambase | 4601 | 57 | 2 | **86.6** | 86.0 | 81.3 | 85.4 | **86.2** | 84.5 | 82.8 | 84.5 |
>  | Wall-Following | 5456 | 2 | 4 | **94.0** | **94.0** | 42.5 | **94.0** | **94.0** | **94.0** | 40.9 | **94.0** |
>  | Page-block | 5473 | 10 | 5 | **95.4** | **95.4** | 93.9 | **95.4** | 95.0 | **95.3** | 92.3 | **95.3** |
>
>  *Here, "RSOCT" represents our proposed method, "Flow" represents the work from [2] and "OCT" represents the work from [6]. All methods are tested after hyperparameter tuning. The detailed tuning procedure can refer to the response under reviewer v4RW.*
>
> Although we have not successfully implemented methods in [1, 3-5] yet, we can still evaluate their performance on continuous datasets using the results reported in these papers. Table 1 in [5] shows that this paper only performs experiments on datasets with less than 1728 samples. Table 1 in [4] shows that the largest dataset used in this paper has 8124 samples. In Table 3 of the paper [3], the authors compare their DL8.5 method [3], and BinOCT proposed in [1] on continuous datasets with 150 to 4521 samples and a max depth of 2 and shows that BinOCT reaches a time-out. The authors also mentioned that "Note that the number of generated features is very high in this case. As a result, for most datasets all algorithms reach a time-out for maximum depths of 3 and 4, as was also shown by Verwer and Zhang (2019). Hence, we focus on results for a depth of 2 in Table 3." We agree with [3] that transforming continuous features into binary could probably generate a huge number of feature space, especially when the unique values of the continuous features are large. For a dataset with S samples and A features, in the worst case, $A\times S$ additional binary features will be introduced if a simple approach [5] is used for binary encoding, or $A\times log(S)$ if a complicated binary encoding is utilized [1].

---

> > ### Author Response · Authors · 2022-08-02
> > **Response to Reviewer fTBj**
> >
> > The above discussion highlights that no other article has ever reported results on continuous datasets of the same scale as our paper. Our code is under continuous development, and the latest version performs much better than our initial submission. The following table shows that now we are addressing the dataset with almost a million samples. After the reviewing process, the code will be released open-source.
> >
> >
> >  Table for Training Optimal Classification Tree with maximum depth=2, with fixed $\alpha=0.05$ on 1000 cores
> >
> >  | Dataset | n | P | K | Time | UB | Gap | Train-CART | Train-RSOCT | Train-uplift | Test-CART | Test-RSOCT | Test-uplift |
> >  |-------------------|--------|----|---|----------|-----------|-------|------------|-------------|--------------|-----------|------------|-------------|
> >  | Sepsis | 110204 | 3 | 2 | 14400.0 | 6559.5 | 4.4% | 92.6 | 92.6 | 0.0% | 92.6 | 92.6 | 0.0% |
> >  | Skin-Segmentation | 245057 | 3 | 2 | 6698.9 | 16953.7 | 1.0% | 90.7 | **92.7** | 2.2% | 90.6 | **92.7** | 2.3% |
> >  | HTS | 928991 | 11 | 3 | 14400.0 | 753075.8 | 56.2% | 58.1 | **59.7** | 2.7% | 58.1 | **59.7** | 2.8% |
> >
> >
> > * **Experiments are limited to trees of depth 2. This is a very shallow depth and many previous works have considered deeper trees.**
> >
> >  Thank you for pointing out this limitation. The following table provides the result of our method for training the optimal classification tree with depth=3.
> >
> >  Table for Training Optimal Classification Tree with maximum depth=3
> >
> >  | Dataset | n | P | K | Train-CART | Train-OCT | Train-Flow | Train-RSOCT | Test-CART | Test-OCT | Test-Flow | Test-RSOCT |
> >  |----------------|------|----|---|------------|-----------|------------|-------------|-----------|-----------|-----------|------------|
> >  | Seeds | 210 | 7 | 3 | 91.8 | **96.9** | 45.3 | 95.0 | **94.1** | 90.2 | 32.5 | **94.1** |
> >  | Glass | 214 | 9 | 6 | 72.2 | **72.8** | 59.2 | 71.5 | 64.3 | **67.9** | 37.0 | 62.5 |
> >  | Body | 507 | 5 | 2 | 91.9 | **93.7** | 63.9 | **93.7** | 88.1 | **93.7** | 59.8 | 91.3 |
> >  | Statlog-German | 1000 | 24 | 2 | 76.7 | 75.5 | **77.2** | 76.0 | **74.8** | 72.4 | 72.0 | 70.8 |
> >  | Concrete | 1030 | 8 | 3 | 69.5 | **69.9** | 66.3 | 67.8 | 65.0 | 64.6 | 65.1 | **70.4** |
> >  | Banknote | 1372 | 4 | 2 | 95.1 | 96.2 | 55.1 | **97.9** | 95.9 | **98.5** | 52.2 | 96.8 |
> >  | Contraceptive | 1473 | 11 | 3 | 53.4 | **55.8** | 53.4 | 55.6 | 52.6 | 56.4 | 46.3 | **56.6** |
> >  | Ozone-eight | 1847 | 72 | 2 | **93.7** | **93.7** | 7.4 | **93.7** | 92.0 | **92.8** | 6.5 | **92.8** |
> >  | Ozone-one | 1848 | 72 | 2 | 97.0 | **97.3** | 96.3 | 97.1 | 96.8 | 96.3 | **97.2** | 96.1 |
> >  | Thyroid-ann | 3772 | 21 | 3 | **99.4** | **99.4** | 93.0 | **99.4** | 99.0 | 98.8 | 94.4 | **99.2** |
> >  | Statlog-lansat | 4435 | 36 | 6 | 79.6 | 79.6 | 33.8 | **80.1** | 79.8 | 79.2 | 33.3 | **79.9** |
> >  | Spambase | 4601 | 57 | 2 | 87.9 | 88.6 | oom | **89.6** | 87.0 | 87.2 | oom | **89.0** |
> >  | Wall-Following | 5456 | 2 | 4 | **100.0** | **100.0** | 43.1 | **100.0** | **100.0** | **100.0** | 41.4 | **100.0** |
> >  | Page-block | 5473 | 10 | 5 | 96.3 | **96.4** | oom | **96.4** | **95.7** | **95.7** | oom | **95.7** |
> >
> >  *Here, "RSOCT" represents our proposed method, "Flow" represents the work from [1] and "OCT" represents the work from [6]. "oom" means the algorithm ran out of memory for an allocation of 16G. All methods are tested after hyperparameter tuning. The detailed tuning procedure can refer to the response under reviewer v4RW*
> >
> >  From the Table, we can see that our method has the ability to train an optimal classification tree with depth=3. The result on most datasets can outperform FlowOCT and CART.
> >
> >  We admit that training a deeper (depth >= 4) decision tree on large datasets with continuous variables is still a challenge. Indeed, current works in the literature about training DT with continuous features all only report results on the shallow decision tree when the dataset is over 1000 samples (see [6] and [7]) (although [6] tested on depth=4, there are almost no improvement on dataset over 1000 samples, compared to CART).

---

> > > ### Author Response · Authors · 2022-08-02
> > > **Response to Reviewer fTBj**
> > >
> > > * **Ablation study: the paper would benefit from an analysis of the impact of different components (e.g., performance without the sample reduction technique).**
> > >
> > >  Thanks for the reviewer's suggestion. In the appendix file from the supplementary material, we provide an analysis of how many samples are determined during the branch and bound calculation. As we can see from the table there, many datasets can almost determine the cost of 90% of samples in some of the branch and bound nodes, which is a tremendous benefit for accelerating the solution time.
> > >
> > >  Following the reviewer's suggestion, we will add more experimental results comparing our method with and without the sample reduction in the final version of the paper. However, due to the limitation of our computing resources, we are still waiting for the resource allocation to execute the ablation study. We hope the experiments can be completed during the discussion period.
> > > * **Minor points:**
> > >
> > >  Thanks a lot for your thorough review of our paper. We will fix all typos and misleading contexts in the revised version.
> > >
> > > [1] Verwer, Sicco, and Yingqian Zhang. "Learning optimal classification trees using a binary linear program formulation." Proceedings of the AAAI conference on artificial intelligence. Vol. 33. No. 01. 2019.
> > >
> > > [2] Aghaei, Sina, Andrés Gómez, and Phebe Vayanos. "Strong optimal classification trees." arXiv preprint arXiv:2103.15965 (2021).
> > >
> > > [3] Aglin, Gaël, Siegfried Nijssen, and Pierre Schaus. "Learning optimal decision trees using caching branch-and-bound search." Proceedings of the AAAI Conference on Artificial Intelligence. Vol. 34. No. 04. 2020.
> > >
> > > [4] Hu, Hao, et al. "Learning optimal decision trees with maxsat and its integration in adaboost." IJCAI-PRICAI 2020, 29th International Joint Conference on Artificial Intelligence and the 17th Pacific Rim International Conference on Artificial Intelligence. 2020.
> > >
> > > [5] Shati, Pouya, Eldan Cohen, and Sheila McIlraith. "SAT-based approach for learning optimal decision trees with non-binary features." 27th International Conference on Principles and Practice of Constraint Programming (CP 2021). Schloss Dagstuhl-Leibniz-Zentrum für Informatik, 2021.
> > >
> > > [6] Bertsimas, D., & Dunn, J. (2017). Optimal classification trees. Machine Learning, 106(7), 1039-1082. ISO 690
> > >
> > > [7] Zhu, H., Murali, P., Phan, D., Nguyen, L., & Kalagnanam, J. (2020). A scalable mip-based method for learning optimal multivariate decision trees. Advances in Neural Information Processing Systems, 33, 1771-1781.

---

> > > > ### Author Response · Authors · 2022-08-08
> > > > **Additional Response to Reviewer fTBj**
> > > >
> > > > We performed additional experiments (we just exhausted our HPC resources:)) and hope the new results can resolve the reviewer's concerns. We are glad to receive the valuable comments, and it is always welcome to ask more questions about our response. We hope that through the rebuttal and discussion, we will convince the reviewer to raise their evaluation. We would really appreciate the support!
> > > >
> > > > * **Experiments on Large Datasets with depth=2 and 3 in Comparison with OCT and Flow.**
> > > >
> > > > The following tables provide the experiments of large datasets (n > 200,000) with depth=2 and 3.
> > > >
> > > > Table for Training Optimal Classification Tree with Maximum Depth=2 on Large Datasets with 1000 Cores
> > > >
> > > > | Dataset | n | P | K | Train-CART | Train-OCT | Train-Flow | Train-RSOCT | Test-CART | Test-OCT | Test-Flow | Test-RSOCT |
> > > > |-------------------|--------|----|---|------------|-----------|------------|-----------|-----------|----------|-----------|----------|
> > > > | Skin-Segmentation | 245057 | 3 | 2 | 90.7 | 90.7 | oom | **92.7** | 90.6 | 90.6 | oom | **92.7** |
> > > > | HTS | 928991 | 11 | 3 | 58.1 | 58.1 | oom | **59.7** | 58.1 | 58.1 | oom | **59.8** |
> > > >
> > > > Table for Training Optimal Classification Tree with Maximum Depth=3 on Large Datasets with 1000 Cores.
> > > >
> > > > | Dataset | n | P | K | Train-CART | Train-OCT | Train-Flow | Train-RSOCT | Test-CART | Test-OCT | Test-Flow | Test-RSOCT |
> > > > |-------------------|--------|----|---|------------|-----------|------------|-----------|-----------|----------|-----------|----------|
> > > > | Skin-Segmentation | 245057 | 3 | 2 | 93.8 | 93.8 | oom | **96.6** | 93.8 | 93.8 | oom | **96.5** |
> > > > | HTS | 928991 | 11 | 3 | 64.3 | 64.3 | oom | **64.6** | 64.3 | 64.3 | oom | **64.5** |
> > > >
> > > > The tables show that the Flow method will fail on all large datasets. Particularly, for the most extensive dataset HTS, following the binary transformation through assigning a single variable to each possible value, it will introduce over 10,000,000 binary features, which makes the Flow method hard to execute. For OCT methods, we notice that after 4-hour execution, OCT can only provide the same solution as CART, which is the warm-start solution provided at the start of the calculation. Our method (RSOCT) reveals robust scalability on large datasets with almost **one million** samples.
> > > >
> > > > * **Ablation Test.**
> > > >
> > > > Here we add the ablation test for the sample reduction. To make the comparison more precise, we only test on datasets where the average percentage of determined samples is above zero (this information can be referred to in Table 3 in Appendix E).
> > > >
> > > > Table for the Ablation Test
> > > >
> > > > | Dataset | n | P | K | Time-No SR | Time-SR | Gap-No SR | Gap-SR |
> > > > |----------------|------|----|---|-------------|------------|-----------|-----------|
> > > > | Seeds | 210 | 7 | 3 | 329.1 | **151.2** | <1% | <1% |
> > > > | Glass | 214 | 9 | 6 | 2,125.6 | **1,903.8**| <1% | <1% |
> > > > | Body | 507 | 5 | 2 | 1,036.1 | **965.9** | <1% | <1% |
> > > > | Banknote | 1372 | 4 | 2 | 773.1 | **633.8** | <1% | <1% |
> > > > | Wall-Following | 5456 | 2 | 4 | 338.3 | **223.2** | <1% | <1% |
> > > > | Contraceptive | 1473 | 11 | 3 | 14,400.0 | 14,400.0 | 12.2% | **12.0%** |
> > > > | Thyroid-ann | 3772 | 21 | 3 | 14,400.0 | 14,400.0 | 68.7% | **55.6%** |
> > > > | Page-block | 5473 | 10 | 5 | 14,400.0 | 14,400.0 | 47.3% | **46.2%** |
> > > >
> > > > *Here, "SR" represents Sample Reduction. Results are obtained with a runtime limit of 4 hours.*
> > > >
> > > > From the table, we can see that, with sample reduction, all datasets can either converge to a smaller optimality gap within the same time or use less time to reach the $\leq$ 1% optimality gap.
> > > >
> > > >
> > > > * **Significance.**
> > > >
> > > > We would like to reiterate that we are the first to provide such a framework for training the optimal decision tree on a large dataset (continuous features with up to almost **1 million** samples) with the following attributes:
> > > >
> > > > 1) We reconstruct the optimal decision tree training as a two-stage stochastic programming problem and provide a reduced-space branch and bound algorithm for this problem.
> > > > 2) We prove that our algorithm can guarantee convergence by only branching on the variables that determine the tree structure (i.e. variable $a,b,c,d$ in Formula (2)).
> > > > 3) Our new MIP formulation consists of a decomposable structure that can be directly parallelized when solving the lower bounding problem. This attribute enables our algorithm to scale to large datasets through trivial parallelism.
> > > > 4) We also proposed the sample reduction method to predetermine the cost of each sample before calculating each node's lower bound.

---

### Official Review · Reviewer_25Mp · 2022-07-12

**Rating:** 5
**Confidence:** 2
**Soundness:** 3 good
**Presentation:** 2 fair
**Contribution:** 3 good

**Summary:**

The paper proposes a branch-and-bound based approach to obtaining the optimal decision tree. The basic idea is to separate the problem into two stages, the first of which is about structure of tree and the second of which is about variables that specifies the allocation and cost of samples.

**Questions:**

In (6) and (10), 'min' should be the inside of \sum? Although the subscript 's' is in the sum index, it is also shown outside of the sum (m_s).

In Algorithm 1, what does 'j \in {1,2} is infeasible' mean?

**Limitations:**

A limitation about the tree depth is described in Section 6.

**Strengths And Weaknesses:**

- The basic idea would be reasonable and the empirical results suggest efficiency of the proposed method.

- In abstract:

> find global optimal solutions with a small optimality gap for datasets with over 245,000 samples.

I feel that this statement is seemingly a bit exaggerated too much because the result is obtained by 1000 cores (though in the introduction, it is described).

- If I correctly understand, the entire general framework (two stage decomposition with branch-and-bound) has been already studies in Cao and Zavala 2019, and Hua et al., 2021. There exist technical differences, but the basic idea of the bound and the algorithm was already established. Although the authors explain it mainly in Introduction, the fact that the proposed method is based on those past work is not fully clear from the descriptions in the paper.

- The variables in (3) are not defined in the main text.

- A (minor) weak point of the paper would be its complexity of the technical writing. Because of the large number of variables and complicated definition of the optimization problem, the paper is a bit difficult to follow. More detailed explanations particularly for the problem (2) would help readers even if it is only in appendix (Actually, I had to check Bertsimas and Dunn 2017 to understand the basic formulation of (2)).

---

> ### Author Response · Authors · 2022-08-02
> **Response to Reviewer 25Mp**
>
> We appreciate the reviewer's clear and insightful comments. Here is our reply to your concerns.
>
> * **Q1 - In abstract: find global optimal solutions with a small optimality gap for datasets with over 245,000 samples. I feel that this statement is seemingly a bit exaggerated too much because the result is obtained by 1000 cores (though in the Introduction, it is described).**
>
>  Thank you for pointing out our immature expression in the abstract. In our original submission, we mentioned the *parallelized bounding strategies* but described the details in the Introduction and Computational Experiments. Following the reviewer's suggestion, we will clarify the setting of the experiments on large datasets (4 hours, 1000 cores, 1.4% optimality gap) in the abstract of the final version.
>
> * **Q2 - If I correctly understand, the entire general framework (two stage decomposition with branch-and-bound) has been already studies in Cao and Zavala 2019, and Hua et al., 2021. There exist technical differences, but the basic idea of the bound and the algorithm was already established. Although the authors explain it mainly in Introduction, the fact that the proposed method is based on those past work is not fully clear from the descriptions in the paper.**
>
>  We apologize for any misleading. In our original submission, we briefly mentioned the framework of [Cao and Zavala, 2019] and [Hua et al., 2021] in the Introduction, while having a detailed discussion on the difference between our work and these papers in Appendix B1 Convergence Analysis (in Supplementary Material). To make it more clear, we will move some discussions to the Introduction of the final version.
>
>  The idea of branching only on the first-stage variables was first proposed in [CarøE and Schultz, 1999] and [Karuppiah and Grossmann, 2008] for stochastic MILPs and stochastic MINLPs, respectively, and the convergence for stochastic NLPs is proved in [Cao and Zavala, 2019]. This idea is recently applied to the K-means clustering problem in [Hua et al., 2021].
>
>  We cannot directly use the general framework of [Cao and Zavala, 2019] and [Hua et al., 2021]. In [Cao and Zavala, 2019], both first and second-stage variables are continuous. They also assume that the optimal value of the second stage problem $Q_s(.)$ is a continuous function. [Hua et al., 2021] proved the convergence for K-means by illustrating that this assumption is satisfied. However, the optimal decision tree problem has mixed-integer first and second-stage variables. Moreover, the $Q_s(.)$ function is discontinuous ( $L_s$ term in $Q_s(.)$ can only be 0 or 1). Therefore, proving the convergence for the optimal decision tree problem (which has integers in **both** the first and second stage variables) is much more complicated.
>
>  Besides being the first to borrow this idea of reduced-space branch and bound from the stochastic programming community and tailor this framework for optimal decision tree problems, we would also like to highlight other contributions. We derived the closed-form solution of the basic lower bound problems, proposed several lower and upper bounding methods, and developed several sample reduction techniques. All of these significantly speed up the solution process and are not mentioned in previous works.
> * **Q3 - Other concerns.**
>  * **The variables in (3) are not defined in the main text.**
>  * **A (minor) weak point of the paper would be its complexity of the technical writing. Because of the large number of variables and complicated definition of the optimization problem, the paper is a bit difficult to follow. More detailed explanations particularly for the problem (2) would help readers even if it is only in appendix (Actually, I had to check Bertsimas and Dunn 2017 to understand the basic formulation of (2)).**
>  * **In (6) and (10), 'min' should be the inside of \sum? Although the subscript 's' is in the sum index, it is also shown outside of the sum (m_s).**
>  * **In Algorithm 1, what does 'j \in {1,2} is infeasible' mean?**
>
>  We apologize for the confusing notations and complicated formulations. We will clarify all the notations as the reviewer suggested. We will also put a detailed explanation of Problem (2) into the appendix of the final version of the paper.

---

### Meta-Review · Area_Chair_ZqLM · 2022-08-26

**Recommendation:** Accept
**Confidence:** Less certain

**Metareview:**

This paper proposes a new approach for computing globally optimal decision trees that is more scalable than prior work. Reviewers gshW, fTBj, and 25Mp are all in favor of accepting the paper, with the main strengths being that:

1. The algorithmic approach is novel and improves scalability of learning globally optimal decision trees.
1. Experiments show that learning globally optimal decision trees improves over baseline methods.
1. The proposed algorithm is empirically shown to be more scalable than prior work.

Reviewer v4RW argues that the paper should be rejected. Their main concern is that the authors select a single regularization parameter for the entire study, seemingly arbitrarily. In response, the authors argue that keeping the regularization parameter fixed when comparing between their proposed method and baselines is to demonstrate the increased accuracy of the learned tree due to optimally solving the optimization problem. They authors also provided new experimental results during the discussion period where the regularization parameter is tuned for each method and dataset. Overall I am convinced by the authors response, and reviewer v4RW did not respond during the discussion period.

The other weaknesses discussed by the reviewers are:

1. Reviewer fTBj argued that the experiments were missing important baselines, experiments were only performed with depth-2 trees, and that the experiments would benefit from an ablation study showing how each component of the proposed algorithm helps performance.  In the discussion period the authors provided results with one additional baseline, trees of depth 3, larger datasets, and an ablation study. Reviewer fTBj was encouraged by these results and raised their score as a result.
1. Reviewer 25Mp asked for clarification on how the proposed method differs from Cao and Zavala (2019) and Hua et al. (2021). In the discussion period, the authors point out a number of technical differences between their work and the prior work, and committed to including this discussion in the paper. I am generally convinced by their arguments and reviewer 25Mp did not respond during the discussion period.

Overall, I feel that the points in favor of the paper outweigh the points against.


**Award:**

No

---

### Decision · Program_Chairs · 2022-09-14

Accept